# SUBZEROCORE: A SUBMODULAR APPROACH WITH ZERO TRAINING FOR CORESET SELECTION

## ABSTRACT

The goal of coreset selection is to identify representative subsets of datasets for efficient model training. Yet, existing approaches paradoxically require expensive training-based signals, e.g., gradients, decision boundary estimates or forgetting counts, computed over the entire dataset prior to pruning, which undermines their very purpose by requiring training on samples they aim to avoid. We introduce Sub-ZeroCore, a novel, training-free coreset selection method that integrates submodular coverage and density into a single, unified objective. To achieve this, we introduce a sampling strategy based on a closed-form solution to optimally balance these objectives, guided by a single hyperparameter that explicitly controls the desired coverage for local density measures. Despite no training, extensive evaluations show that SubZeroCore matches training-based baselines and significantly outperforms them at high pruning rates, while dramatically reducing computational overhead. SubZeroCore also demonstrates superior robustness to label noise, highlighting its practical effectiveness and scalability for real-world scenarios. Our code is publicly available at `https://github.com/WILL-BE-IN-FINAL/subzerocore`.

## 1 INTRODUCTION

Deep learning breakthroughs often stem from training ever-larger models on ever-larger datasets, a trend that is both resource-heavy and environmentally costly Wang et al. (2018); Csiba & Richtárik (2018); Zheng et al. (2022); Katharopoulos & Fleuret (2018). In many applications, however, collecting or storing vast amounts of data poses significant challenges Ganguli et al. (2022); Yang & Su (2024). Coreset selection seeks to address these problems by identifying a subset that contains a sufficient yet representative data summary of the original dataset Moser et al. (2025); Sorscher et al. (2022); Guo et al. (2022). In principle, such a coreset, once found, allows one to train models more efficiently on a fraction of the data without sacrificing much training quality Katharopoulos & Fleuret (2018); Bhalerao (2024). Sometimes, they even lead to better training performance by mitigating the risk of injecting poisoned data into training, i.e., data with noisy annotations or outliers Katharopoulos & Fleuret (2018); Bengio et al. (2019); Marion et al. (2023); Ren et al. (2018). Examples of such positive effects can be found in various deep learning fields like neural architecture search Na et al. (2021); Moser et al. (2022); Yao et al. (2023), image enhancement Moser et al. (2024a); Ding et al. (2023); Laribi et al. (2024), dataset distillation Moser et al. (2024b); Chen et al. (2024); Khandel et al. (2024), imbalanced datasets Sivasubramanian et al. (2024); Luo et al. (2024), continual learning Nguyen et al. (2017); Borsos et al. (2020); Yoon et al. (2021), and even quantum machine learning Qu et al. (2022); Huang et al. (2024); Xue et al. (2023).

An ideal coreset selection method must balance two competing goals: **coverage**, which measures how well a selected subset represents the overall diversity and distribution of the full dataset, and **density**, which identifies highly concentrated regions in the data space containing informative, but potentially redundant samples Zheng et al. (2022); Sener & Savarese (2017); Koh & Liang (2017). Despite recent progress, state-of-the-art methods often incur heavy computational overhead because they rely on training-based signals such as gradients Paul et al. (2021); Mirzasoleiman et al. (2020); Killamsetty et al. (2021a), forgetting scores Toneva et al. (2018); Paul et al. (2021), or decision boundary estimates Ducoffe & Precioso (2018); Margatina et al. (2021). While these signals can help to identify impactful samples, they require partial or complete model training and also subject to exhaustive hyperparameter search Guo et al. (2022). Paradoxically, this means current coreset selection methods, intended to reduce training burdens, often require extensive training and evaluations themselves.

In this work, we propose **SubZeroCore**, a novel coreset selection method grounded in submodular optimization Bérczi et al. (2019) that requires *zero model training*. Unlike existing gradient-based or loss-dynamic methods, SubZeroCore uniquely integrates both coverage and density into a single, submodular objective. As such, SubZeroCore positions itself among geometry-based methods like k-center greedy but with an objective for optimizing density as well as coverage. By leveraging a closed-form coverage estimate to compute a hyperparameter-efficient local density, our method systematically picks a suitable neighborhood size with no reliance on gradients or iterative training. The result is a coreset selection method that (i) avoids expensive model-specific signals, (ii) maintains high coverage but still focuses on dense regions, (iii) offers theoretical optimality guarantees through submodularity, and (iv) relies on a single, controllable hyperparameter.

Concretely, we demonstrate that our submodular objective captures both coverage and density to improve the quality of coreset. Our experiments on CIFAR-10 Krizhevsky et al. (2009) as well as ImageNet-1K Deng et al. (2009) show that SubZeroCore consistently performs comparable to training-based baselines for low pruning rates and outperforms them under high pruning rates, while being substantially faster than most training-based approaches. Moreover, as emphasizing dense regions naturally de-emphasizes outliers, SubZeroCore remains robust to label noise.

Taken together, our findings frame SubZeroCore as a practical tool for scalable coreset selection. We believe this approach offers a practical avenue for advancing coreset-based strategies in domains where data curation or resource constraints predominate Lee et al. (2021); Abbas et al. (2024).

## 2 PRELIMINARIES

### 2.1 CORESET SELECTION

We begin with a classical discriminative task, where the training dataset $\mathcal{T} = \{(\mathbf{x}_i, y_i)\}_{i=1}^N$ consists of $N$ i.i.d. samples drawn from an underlying data distribution $P$. Each input $\mathbf{x}_i \in \mathcal{X}$ is paired with a ground-truth label $y_i \in \mathcal{Y}$.

**Definition 1** (Coreset Selection). *The goal of coreset selection is to derive a small subset $\mathcal{S} \subset \mathcal{T}$ ($|\mathcal{S}| \ll |\mathcal{T}|$) such that training a model $\theta^{\mathcal{S}}$ on $\mathcal{S}$ yields generalization performance on par with $\theta^{\mathcal{T}}$ trained on the entire dataset $\mathcal{T}$:*

$$\mathcal{S}^* = \underset{\mathcal{S} \subset \mathcal{T}: \frac{|\mathcal{S}|}{|\mathcal{T}|} \approx 1-\alpha}{\arg\min} \quad \mathbb{E}_{\mathbf{x}, y \sim P}\big[\mathcal{L}\big(\mathbf{x}, y; \theta^{\mathcal{S}}\big) - \mathcal{L}\big(\mathbf{x}, y; \theta^{\mathcal{T}}\big)\big], \tag{1}$$

*where $\alpha \in (0, 1)$ is the pruning ratio (fraction of samples removed) and $\mathcal{L}$ is a loss function.*

While this objective is conceptually straightforward, it can be difficult to realize in practice Agarwal et al. (2005); Feldman (2020); Bachem et al. (2017). One must decide how best to measure "importance" or "representativeness" for each sample $\mathbf{x}_i$, so that the selection algorithm can prioritize those samples that most benefit the training Nogueira et al. (2018); Song et al. (2022); Xiao et al. (2025); Swayamdipta et al. (2020). For the remainder of this work, we focus on class-wise selection algorithms. Accordingly, we adopt the simplified notation $\mathbf{x} \sim P$ instead of $(\mathbf{x}, y) \sim P$. Thus, we also denote datasets using the simplified notation $\mathcal{T} = \{\mathbf{x}_i\}_{i=1}^N$ and selected coresets as $\mathcal{S} = \{\mathbf{x}_i\}_{i=1}^{(1-\alpha)\cdot N}$.

### 2.2 SUBMODULAR FUNCTIONS

Submodularity is a fundamental property of set functions that captures the principle of diminishing returns. Since we are interested in selecting the most informative samples first, the submodularity property is especially attractive for coreset selection Iyer & Bilmes (2013); Kothawade et al. (2021); Karanam et al. (2022); Wei et al. (2015); Dou et al. (2023).

**Definition 2** (Submodularity). *A function $f : 2^V \to \mathbb{R}$ defined over a ground set $V$ is called submodular if, for any subsets $A \subseteq B \subseteq V$ and any element $j \in V \setminus B$, it holds that*

$$f(A \cup \{j\}) - f(A) \geq f(B \cup \{j\}) - f(B). \tag{2}$$

*This diminishing-returns condition intuitively says that adding an element to a smaller set provides a larger marginal gain than adding it to a bigger set Iyer et al. (2021).*

Formally, coreset selection can be posed as maximizing a submodular function under a budget constraint:

$$\mathcal{S}^* = \underset{\mathcal{S} \subset \mathcal{T}:\ \frac{|\mathcal{S}|}{|\mathcal{T}|} \approx 1-\alpha}{\arg\max} f(\mathcal{S}), \qquad (3)$$

where $f$ is submodular, $\mathcal{T}$ indexes all data samples, and $\alpha$ is the pruning factor. A common submodular example is facility location.

**Definition 3** (Facility Location). *Facility location Bérczi et al. (2019); Wei et al. (2014) defines a submodular function $f_{FL} : 2^{\mathcal{T}} \to \mathbb{R}$:*

$$f_{FL}(\mathcal{S}) = \sum_{\mathbf{x} \in \mathcal{T}} \max_{\mathbf{x}_S \in \mathcal{S}} \text{sim}(\mathbf{x}, \mathbf{x}_S), \qquad (4)$$

*where* sim *is typically a similarity function (e.g., cosine) Iyer & Bilmes (2013). The facility location function inherently favors coverage because it evaluates each data sample in the entire dataset by taking the maximum similarity to any sample in the selected subset.*

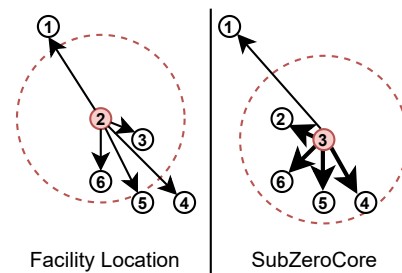

Figure 1: The goal of SubZeroCore: In addition to maximizing coverage through facility location **(left)**, SubZeroCore **(right)** also incorporates a density-based weighting scheme, which prioritizes selecting data samples (red dots) from regions of higher local density as emphasized by the bold arrows.

Although finding the exact optimal subset $\mathcal{S}^*$ under a submodular objective $f$ is generally NP-hard Svitkina & Fleischer (2011); Iyer et al. (2013), submodular functions enjoy a crucial advantage: they can be approximately maximized via a simple greedy algorithm. For the cardinality-constrained case (i.e., limited subset size), the classical result by *Nemhauser et al.* Nemhauser et al. (1978) guarantees that greedy selection achieves a $(1 - 1/e) \approx 63\%$ approximation ratio: $f(\mathcal{S}_{\text{greedy}}) \geq (1 - 1/e) f(\mathcal{S}^*)$. This tells us that (i) greedy selection obtains a strong approximation without exhaustive search, (ii) the greedy algorithm guarantees to achieve at least about 63% of the maximum possible score of the chosen submodular function (such as facility location), and (iii) lazy-greedy optimizations Lim et al. (2014); Lundberg & Lee (2017) can reduce computational cost significantly. While one might ask "Why not 100%?", the answer is that each greedy step picks the locally best option at that moment, without accounting for future interactions among samples. Yet, *greedy suboptimality* has been a well-understood limitation in submodular maximization since 1978, but in practice, the $(1 - 1/e)$ bound on the submodular metric score is often considered both strong and acceptable Bérczi et al. (2019); Nemhauser et al. (1978); Lim et al. (2014).

## 3 METHODOLOGY

The goal of coreset selection is to select data samples that **(i)** collectively achieve sufficient coverage of the underlying data distribution and **(ii)** lie in high-density regions. Since both objectives usually counteract each other, existing methods generally choose just one objective: However, for high pruning ratios, one desires a high-density driven coreset selection method, while a coverage-based method is more favorable for low pruning ratios Zheng et al. (2022); Sener & Savarese (2017).

Thus, balancing both density and coverage within a unified framework remains a significant yet challenging objective. We propose SubZeroCore, a new method that combines submodular optimization, i.e, facility location-based coverage maximization, with density-driven importance weighting, as illustrated in Figure 1. The complete algorithm can be found in the appendix.

### 3.1 CONCERNING DENSITY

**Definition 4** (Density). *For a data sample $\mathbf{x}$, we define its density by finding the size of the neighborhood needed to capture $K$ nearest neighbors in $\mathcal{T}$. If we define the radius by $r = \text{NND}_K(\mathbf{x})$, where $\text{NND}_K(\mathbf{x})$ denotes the distance of $\mathbf{x}$ to its $K$-th nearest-neighbor, then a common way to express density $\rho_K : \mathcal{T} \to [0, \infty]$ is via*

$$\rho_K(\mathbf{x}) = \frac{|B(\mathbf{x}, r) \cap \mathcal{T}|}{\text{Vol}(B(\mathbf{x}, r))}, \qquad (5)$$

*where $B(\mathbf{x}, r)$ is a ball around $\mathbf{x}$ with radius $r$, Vol is the volume Morgan (2016), and $|B(\mathbf{x}, r) \cap \mathcal{T}|$ are the amount of elements in $\mathcal{T}$ within that ball. Note that $|B(\mathbf{x}, r) \cap \mathcal{T}| > K$ can occur when there are multiple neighbors with exactly $\text{NND}_K(\mathbf{x})$ distance to $\mathbf{x}$ (also exemplified later in Figure 2).*

Informally, density measures how crowded or populated the local region is, thus high for samples with strong support in the real dataset. For further simplifications, we introduce the following lemma:

**Lemma 1.** *For a given $K$ and any two samples $\mathbf{x}_i, \mathbf{x}_j \in \mathcal{T}$ it holds that $\mathrm{NND}_K(\mathbf{x}_i) < \mathrm{NND}_K(\mathbf{x}_j) \Leftrightarrow \rho_K(\mathbf{x}_i) > \rho_K(\mathbf{x}_j)$. In other words, a sample that requires a smaller radius to capture K neighbors is in a denser region.*

*Proof.* Consider $r_{\mathbf{x}_i} = \mathrm{NND}_K(\mathbf{x}_i)$ and $r_{\mathbf{x}_j} = \mathrm{NND}_K(\mathbf{x}_j)$ such that $r_{\mathbf{x}_i} < r_{\mathbf{x}_j}$. Since the volume Morgan (2016) of a ball in a $d$-dimensional metric space

$$\mathrm{Vol}\left(B(\mathbf{x}, r)\right) = \frac{\pi^{\frac{d}{2}}}{\Gamma\left(\frac{d}{2} + 1\right)} r^d$$

is strictly increasing with respect to its radius, it trivially follows that $\mathrm{Vol}\left(B(\mathbf{x}_i, r_{\mathbf{x}_i})\right) < \mathrm{Vol}\left(B(\mathbf{x}_j, r_{\mathbf{x}_j})\right) \Leftrightarrow \rho_K(\mathbf{x}_i) > \rho_K(\mathbf{x}_j)$. ∎

Consequently, the ordering of density for each individual sample $\mathbf{x}_i$ depends by how large or small its ball radius $\mathrm{NND}_K(\mathbf{x}_i)) = r_i$ is compared to other samples : (1) If the radius $r_i$ is small, the sample $\mathbf{x}_i$ lies in a densely populated region because its closest neighbors are spatially closer to it. (2) If the radius $r_i$ is large, the sample $\mathbf{x}_i$ lies in a sparsely populated region, implying fewer samples within close proximity.

## 3.2 SUBZEROCORE

By integrating the density measure for a single sample as a weighting to the facility location, which maximizes coverage, we straightforwardly derive a submodular function dubbed **SubZeroCore** that encourages both aspects, namely coverage and density.

**Definition 5** (SubZeroCore). *Given a data sample $\mathbf{x} \in \mathcal{T}$, we define its density based on Equation 5 by comparing its radius to the overall distribution of neighborhood radii. Simply put, a smaller radius implies higher density (see Lemma 1). More formally, let $\{r_i\}_{i=1}^{|\mathcal{T}|}$ be the radii derived from a fixed K via $r_i = \mathrm{NND}_K(\mathbf{x}_i)$. We compute a sample density score by its relation to the empirical mean $\mu = \frac{1}{|\mathcal{T}|} \sum_{i=1}^{|\mathcal{T}|} r_i$ and standard deviation $\sigma$ of the radii distribution. We then define a density score*

$$s_i = \exp\left(-\frac{(r_i - \mu)^2}{2\,\sigma^2}\right). \tag{6}$$

*By using this normalization, we ensure that density scores are smoothly and consistently assigned, with the highest scores centered around samples whose radii are close to the average density $\mu$, clearly highlighting average dense regions and systematically down-weighting sparse outliers or overly dense inliers. We then feed these density scores into a weighted facility location function $f_{SubZeroCore} : 2^{\mathcal{T}} \to \mathbb{R}$:*

$$f_{SubZeroCore}(\mathcal{S}) = \sum_{\mathbf{x}_i \in \mathcal{T}} \max_{\mathbf{x}_j \in \mathcal{S}} \left(s_j \cdot \mathrm{sim}(\mathbf{x}_i, \mathbf{x}_j)\right), \tag{7}$$

*where $\mathcal{T}$ indexes the entire set of samples in a class, $\mathcal{S} \subseteq \mathcal{T}$ denotes a candidate coreset, and $\mathrm{sim}(\mathbf{x}_i, \mathbf{x}_j)$ is, for instance, a cosine similarity defined on the embeddings of $\mathbf{x}_i$ and $\mathbf{x}_j$. The term $s_j$ acts as a density-based weight emphasizing samples in averagely crowded regions.*

**Corollary 1.** *The SubZeroCore function $f_{SubZeroCore}$ is submodular.*

*Proof.* This directly follows from *Berczi et al.* Bérczi et al. (2019) and can be found in the appendix. ∎

## 3.3 IMPACT OF THE RADIUS AND ITS COVERAGE

**Radius.** The notion of density in dataset pruning heavily relies on the selection of $K$, which sets the scale at which we measure local density, as shown in Figure 2. This is due to the fact that the volume is monotonically increasing with increasing $K$ and from $B(\mathbf{x}, \mathrm{NND}_K(\mathbf{x})) \subseteq B(\mathbf{x}, \mathrm{NND}_{K+1}(\mathbf{x}))$. Consequently, if $K$ is small, density estimates become overly sensitive to isolated samples (overfitting outliers). Conversely, too large $K$ smooths density differences.

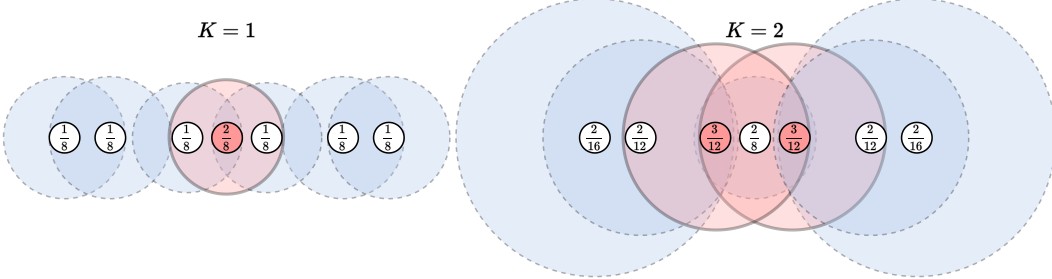

Figure 2: Visualization of how the notion of sample density, defined as the number of neighbors divided by the volume (see numbers in circles), varies depending on the chosen hyperparameter $K$. Red indicates the densest samples for each setting of $K$. As $K$ increases, the density changes and, more importantly, so does the ordering.

Unfortunately, a balanced selection of $K$ depends on the size of the underlying dataset $\mathcal{T}$ and the pruning ratio $\alpha$. To address this, we directly tie $K$ to an interpretable, desired coverage target $\gamma$ between 0% and 100% for the density calculation, thereby systematically guiding the scale at which our method optimally balances coverage with density.

**Coverage.** Inspired by the image synthesis domain and *Naeem et al.* Naeem et al. (2020), we define:

**Definition 6** (Coverage). *Coverage is a measure for what fraction of $\mathcal{T}$-neighborhoods contain a sample of the coreset $\mathcal{S}$. More formally,*

$$\text{coverage}_K(\mathcal{S}, \mathcal{T}) := \frac{1}{|\mathcal{T}|} \sum_{\mathbf{x} \in \mathcal{T}} \mathbf{1}_{\exists\, \mathbf{x}_\mathcal{S} \,\in\, \mathcal{S}\ s.t.\ \mathbf{x}_\mathcal{S} \in B(\mathbf{x}, \text{NND}_K(\mathbf{x}))}. \tag{8}$$

*where $B(\mathbf{x}, \text{NND}_K(\mathbf{x}))$ is again a ball around $\mathbf{x}$ with radius $\text{NND}_K(\mathbf{x})$, which is defined by its distance to its $K$-th nearest-neighbor.*

**Lemma 2.** *The expected coverage of a coreset of size $s \leq |\mathcal{T}| - K$ and a given $K$ is*

$$\mathbb{E}_{\mathcal{S} \sim \mathcal{U}(2^\mathcal{T})}\big[\text{coverage}_K(\mathcal{S}, \mathcal{T})\big||\mathcal{S}| = s\big] = 1 - \prod_{k=0}^{K} \frac{(|\mathcal{T}| - s - k)}{(|\mathcal{T}| - k)}. \tag{9}$$

*Proof.*

$$\mathbb{E}_{\mathcal{S} \sim \mathcal{U}(2^\mathcal{T})}\big[\text{coverage}_K(\mathcal{S}, \mathcal{T})\big||\mathcal{S}| = s\big] = \frac{1}{|\mathcal{T}|} \sum_{\mathbf{x} \in \mathcal{T}} \mathbb{P}\left[\exists\, \mathbf{x}_\mathcal{S} \in \mathcal{S}\ \text{s.t.}\ \mathbf{x}_\mathcal{S} \in B(\mathbf{x}, \text{NND}_K(\mathbf{x}))\right]$$

$$\overset{(i)}{=} 1 - \mathbb{P}\left[\forall\, \mathbf{x}_\mathcal{S} \in \mathcal{S},\ \mathbf{x}_\mathcal{S} \notin B(\mathbf{x}_1, \text{NND}_K(\mathbf{x}_1))\right]$$

$$= 1 - \mathbb{P}\left[S \cap B(\mathbf{x}_1, \text{NND}_K(\mathbf{x}_1)) = \emptyset\right]$$

Since by the uniform nature of $\mathcal{S}$ all samples $x \in \mathcal{T}$ are treated equally, we can fix a particular test sample $\mathbf{x}_1 \in \mathcal{T}$ in step (i). The notation $\mathbf{x}_1$ emphasizes that this sample is now held fixed when we compute $\mathbb{P}[\forall\, \mathbf{x}_\mathcal{S} \in \mathcal{S} : \ldots]$. It plays the same role as any other $\mathbf{x}$ in $\mathcal{T}$. We can reformulate the probability as follows:

> Let $Z = (z_1, \ldots, z_{|\mathcal{T}|})$ be $|\mathcal{T}|$ non-negative real numbers distributed i.i.d. according to $\mathbb{P}_Z$. Select $|\mathcal{S}| = s$ many of them uniformly at random, i.e., the expected value is over $\mathcal{S} \sim \mathcal{U}(2^\mathcal{T})$. What is the probability that the $K$ smallest entries among $Z$ are not in $\mathcal{S}$?

Since the selection is equally likely, we can calculate the probability by counting the ratio of possible selections where the $K$ smallest elements are not selected, which for $|\mathcal{S}| < |\mathcal{T}| - K$ boils down to:

$$\mathbb{P}\left[\forall\, \mathbf{x}_\mathcal{S} \in \mathcal{S},\ \mathbf{x}_\mathcal{S} \notin B(\mathbf{x}_1, \text{NND}_K(\mathbf{x}_1))\right] = \frac{\binom{|\mathcal{T}| - K}{|\mathcal{S}|}}{\binom{|\mathcal{T}|}{|\mathcal{S}|}} = \prod_{k=0}^{K} \frac{(|\mathcal{T}| - |\mathcal{S}| - k)}{(|\mathcal{T}| - k)}.$$

For $|\mathcal{S}| \geq |\mathcal{T}| - K$ (not interesting for coreset selection), the probability becomes 1. $\blacksquare$

### 3.4 Determining the Radius

Figure 3 illustrates how the expected coverage (Equation 9) evolves as $K$ increases under varying pruning levels. We see that the expected coverage tends to rise concavely, indicating diminishing returns once a sufficiently large neighborhood is considered. Higher pruning ratios accentuate this effect, as removing more samples reduces the coverage for a given radius-defining $K$.

Following this analysis, we repurpose the closed-form expectation in Equation 9 to estimate a suitable value of $K$ for our density calculation under a target coverage. Concretely, for a given coverage goal $\gamma \in (0, 1)$, one can (numerically) invert the expression for assigning $K$ to

$$\min\left\{ K \in \mathbb{N} \middle| 1 - \gamma \le \prod_{k=0}^{K} \frac{(|\mathcal{T}| - |\mathcal{S}| - k)}{(|\mathcal{T}| - k)} \right\}$$

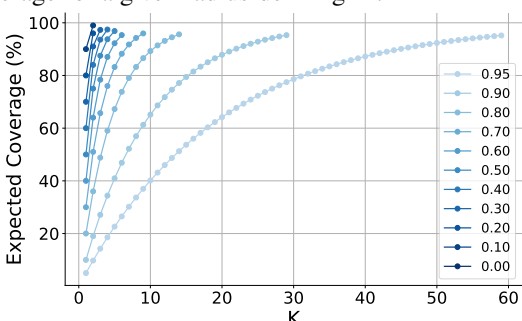

Figure 3: Expected coverage as a function of $K$ across varying pruning ratios. As $K$ increases, the expected coverage follows a nonlinear trajectory, aligning with the expectation of diminishing returns of additional samples under pruning.

which finds a suitable $K$ that achieves $\text{coverage}_K(\mathcal{S}, \mathcal{T}) \approx \gamma$ under the given conditions. Once this $K$ is determined, we can substitute it back into the density formula in Equation 7 to assign an importance weight to each sample. More details in the appendix.

Notably, the expected coverage in Equation 9 is agnostic to the underlying data and coreset distribution, which means we can calculate it without requiring any training or knowledge about the dataset except its magnitude. In other words, the distance-based counting of neighbors in the set $\mathcal{S}$ (scaled by the chosen $K$) provides a straightforward training-free importance weighting scheme. This ensures that samples that are more densely surrounded receive greater importance in subsequent pruning.

In summary, by estimating and settling on such a $K$, we unify coverage and density into a single selection procedure. Specifically, once $K$ is determined from our coverage objective (Equation 9), we compute the $K$-nearest-neighbor radii for each data sample $\mathbf{x}_i$. We then greedily select from $\mathcal{T}$ the subset $\mathcal{S}$ of the required size $|\mathcal{S}| = (1 - \alpha) \cdot |\mathcal{T}|$ that maximizes $f_{\text{SubZeroCore}}(\mathcal{S})$ in Equation 7. Owing to the submodularity and monotonicity of the facility location objective, this greedy selection achieves the $(1 - 1/e)$ approximation guarantee (see *Nemhauser et al.* Nemhauser et al. (1978)).

Overall, SubZeroCore systematically and effectively reconciles the often competing demands of coverage and density within a single submodular optimization target. By deriving the single hyperparameter $K$ from a closed-form solution, our method achieves a robust and efficient coreset selection without any training overhead, making it practically attractive for scalable deep-learning applications.

### 3.5 Implications for Submodularity and Global Coverage

Since the density scores $s_j$ are smaller in sparse regions (where $r_j$ is large) and close to 1 in averagely denser regions (where $r_j$ is small), the weighted objective penalizes the contribution of samples in sparse areas even if they might improve global coverage. Thus, for lower pruning ratios and smaller $K$, our approach tends to

Table 1: Coverage on CIFAR-10 calculated with respect to the corresponding $K$-value: Equation 9 with target coverage $\gamma = 0.6$ delivered the $K$-values 84, 18, 9, 3 for pruning factors 99%, 95%, 90%, 70%, respectively.

| Pruning Factor ($\alpha$) | 99% | 95% | 90% | 70% |
|---|---|---|---|---|
| Facility Location | **56.16** | 60.36 | 60.97 | 65.38 |
| **SubZeroCore (ours)** | 46.77 | **73.73** | **80.49** | **89.03** |

lead to better coreset coverage due to its focus on averagely dense regions, while for higher pruning ratios and higher $K$, it tends to lead to lower coreset coverage. As a consequence, its focus on data density over data coverage is more profound for high pruning ratios, a property generally favorable for coreset selection Sener & Savarese (2017). Empirical validation is provided in Table 1.

Table 2: Coreset performances on CIFAR-10 with five randomly initialized ResNet-18 He et al. (2016) models. Without pruning ($\alpha = 0\%$), the model reaches 95.6±0.1.

| Pruning Factor ($\alpha$) | 99.9% | 99.5% | 99% | 95% | 90% | 80% | 70% | 60% | 50% | 40% | 10% | Train Signals |
|---|---|---|---|---|---|---|---|---|---|---|---|---|
| Random | 21.0±0.3 | 30.8±0.6 | 36.7±1.7 | 62.5±1.1 | 75.7±2.0 | 87.1±0.5 | 90.2±0.3 | 92.1±0.1 | 93.3±0.2 | 94.0±0.2 | 95.2±0.1 | ✗ |
| Herding Welling (2009) | 19.8±2.7 | 29.2±2.4 | 31.1±2.9 | 50.7±1.6 | 63.1±3.4 | 75.2±1.0 | 80.8±1.5 | 85.4±1.2 | 88.4±0.6 | 90.9±0.4 | 94.4±0.1 | ✗ |
| k-Center Greedy Sener & Savarese (2017) | 19.9±0.9 | 25.3±0.9 | 32.6±1.6 | 55.6±2.8 | 74.6±0.9 | **87.3±0.2** | 91.0±0.3 | 92.6±0.2 | 93.5±0.5 | 94.3±0.2 | 95.5±0.2 | ✗ |
| Forgetting Toneva et al. (2018) | 21.3±1.2 | 29.7±0.3 | 35.6±1.0 | 51.1±2.0 | 66.9±2.0 | 86.6±1.0 | **91.7±0.3** | **93.0±0.2** | 94.1±0.2 | 94.6±0.2 | 95.4±0.1 | ✓ |
| GraNd Paul et al. (2021) | 14.6±0.8 | 17.2±0.8 | 18.6±0.8 | 28.9±0.5 | 41.3±1.3 | 71.1±1.3 | 88.3±1.0 | **93.0±0.4** | **94.8±0.1** | **95.2±0.1** | 95.5±0.1 | ✓ |
| CCS (Gradient) Zheng et al. (2022) | 19.1±2.2 | 29.2±2.0 | 36.5±1.1 | 62.8±2.6 | 73.1±0.8 | 86.3±0.2 | 89.9±0.2 | 90.0±0.1 | 90.0±0.1 | 89.9±0.2 | 90.0±0.2 | ✓ |
| ELFS Zheng et al. (2025) | 13.7±0.7 | 20.9±1.0 | 25.3±1.1 | 39.7±1.1 | 52.7±1.9 | 76.8±2.5 | 89.2±0.4 | 91.7±0.3 | 91.9±0.1 | 92.3±0.2 | 92.6±0.1 | ✓ |
| CAL Margatina et al. (2021) | 23.1±1.8 | 31.7±0.9 | 39.7±3.8 | 60.8±1.4 | 69.7±0.8 | 79.4±0.9 | 85.1±0.7 | 87.6±0.3 | 89.6±0.4 | 90.9±0.4 | 94.7±0.2 | ✓ |
| DeepFool Ducoffe & Precioso (2018) | 18.7±0.9 | 26.4±1.1 | 28.3±0.6 | 47.7±3.5 | 61.2±2.8 | 82.7±0.5 | 90.8±0.5 | 92.9±0.2 | 94.4±0.1 | 94.8±0.1 | **95.6±0.1** | ✓ |
| Craig Mirzasoleiman et al. (2020) | 19.3±0.3 | 29.1±1.6 | 32.8±1.8 | 42.5±1.7 | 59.9±2.1 | 78.1±2.5 | 90.0±0.5 | 92.8±0.2 | 94.3±0.2 | 94.8±0.1 | 95.5±0.1 | ✓ |
| GradMatch Killamsetty et al. (2021a) | 17.4±1.6 | 27.1±1.1 | 27.7±2.0 | 41.8±2.4 | 55.5±2.3 | 78.1±2.0 | 89.6±0.7 | 92.7±0.5 | 94.1±0.2 | 94.7±0.3 | 95.4±0.1 | ✓ |
| Glister Killamsetty et al. (2021b) | 18.4±1.3 | 26.5±0.7 | 29.4±1.9 | 42.1±1.0 | 56.8±1.8 | 77.2±2.4 | 88.8±0.6 | 92.7±0.4 | 94.2±0.1 | 94.8±0.2 | 95.5±0.1 | ✓ |
| TDDS Zhang et al. (2024) | 18.3±1.0 | 32.4±0.9 | 39.1±1.1 | 63.7±1.1 | 76.8±1.7 | 87.1±0.3 | 90.6±0.4 | 92.5±0.1 | 93.3±0.0 | 94.0±0.2 | 95.3±0.1 | ✓ |
| Facility Location | 21.0±1.3 | 30.3±1.2 | 38.1±1.3 | 58.8±2.3 | 70.9±1.9 | 86.6±0.9 | 91.2±0.4 | 92.9±0.2 | 94.3±0.1 | 94.7±0.1 | 95.5±0.1 | ✗ |
| **SubZeroCore (ours)** | **24.0±1.9** | **32.9±1.5** | **39.8±1.1** | **63.9±2.0** | **77.4±0.8** | 87.3±0.5 | 90.8±0.3 | 92.5±0.1 | 93.2±0.1 | 94.1±0.1 | 95.3±0.1 | ✗ |

Table 3: Comparison against InfoMax Tan et al. (2025), an extension of D2Pruning Maharana et al. (2024), with Forgetting, EL2N, and Entropy scoring on CIFAR-10 with five randomly initialized ResNet-18 He et al. (2016) models. Without pruning ($\alpha = 0\%$), the model reaches 95.6±0.1.

| Pruning Factor ($\alpha$) | 99.9% | 99.5% | 99% | 95% | 90% | 80% | 70% | 60% | 50% | 40% | 10% | Train Signals |
|---|---|---|---|---|---|---|---|---|---|---|---|---|
| Random | 21.0±0.3 | 30.8±0.6 | 36.7±1.7 | 62.5±1.1 | 75.7±2.0 | 87.1±0.5 | 90.2±0.3 | 92.1±0.1 | **93.3±0.2** | 94.0±0.2 | 95.2±0.1 | ✗ |
| Forgetting Toneva et al. (2018) | 12.4±0.4 | 15.6±0.3 | 19.9±0.8 | 33.8±0.8 | 57.1±2.7 | 84.0±0.7 | 87.8±0.2 | 89.7±0.2 | 91.3±0.2 | 92.5±0.4 | 94.9±0.1 | ✓ |
| EL2N Paul et al. (2021) | 11.0±0.2 | 10.7±0.4 | 12.5±0.5 | 23.5±0.3 | 55.6±4.9 | 86.5±0.6 | 88.9±0.3 | 90.5±0.2 | 91.5±0.3 | 92.4±0.1 | 95.0±0.2 | ✓ |
| Entropy | 18.1±0.5 | 25.6±0.8 | 33.1±0.5 | 55.8±3.7 | 71.8±0.2 | 86.7±0.4 | 89.0±0.5 | 90.7±0.3 | 91.6±0.2 | 92.6±0.1 | 95.1±0.1 | ✓ |
| **SubZeroCore (ours)** | **24.0±1.9** | **32.9±1.5** | **39.8±1.1** | **63.9±2.0** | **77.4±0.8** | **87.3±0.5** | **90.8±0.3** | **92.5±0.1** | 93.2±0.1 | **94.1±0.1** | **95.3±0.1** | ✗ |

### 3.6 CLASS-WISE PARTITIONING AND LABEL USAGE

SubZeroCore does not use class labels when computing density scores, similarities, or the submodular objective. All scoring and selection steps operate purely on the embedding geometry. However, for efficiency, the dataset is partitioned class-wise, and the selection procedure is applied independently within each class. Because SubZeroCore selects independent and fixed quotas within each class, minority classes are not overshadowed by majority classes during scoring or selection. This makes the method naturally robust to class imbalance, in contrast to training-signal-based methods whose behavior can shift with changes in class frequency.

## 4 EXPERIMENTS

This section provides our experiments on CIFAR-10 Krizhevsky et al. (2009) and ImageNet-1K Deng et al. (2009), which evaluates our method SubZeroCore under various aspects, such as overall coreset quality, runtime, and robustness.

### 4.1 CIFAR-10 RESULTS

**Setup.** For CIFAR-10, we follow the training protocols of DeepCore Guo et al. (2022). Concretely, we use five ResNet-18 He et al. (2016) models trained with stochastic gradient descent (SGD) on coresets for 200 epochs, using a batch size of 128, an initial learning rate of 0.1 with cosine annealing, momentum 0.9, and weight decay $5 \times 10^{-4}$ and evaluate the trained model on the standard CIFAR-10 test set. We subselect multiple fractions from the full training set, whose performance we treat as an approximate upper bound. Data augmentation includes a random 4-pixel padding followed by cropping to $32 \times 32$, and random horizontal flips.

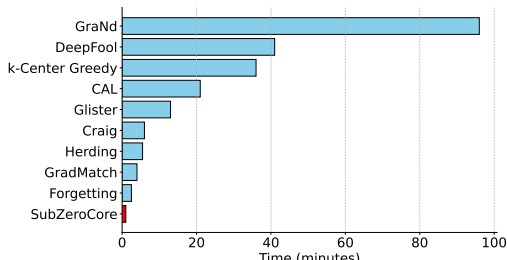

Figure 4: Time-Measurement on CIFAR-10. The bar chart compares the selection times (in minutes) of various methods ($\alpha = 0.99$). SubZeroCore (red) significantly outperforms all other methods, requiring only 1 minute, while other techniques take substantially longer due to the prior training phase before pruning.

Table 4: Coreset selection performances on ImageNet-1K. We train randomly initialized ResNet-18 on the pruned subsets produced by various methods and test on the real ImageNet test set. DeepFool and GraNd were omitted due to their significant memory requirements and runtime.

| Pruning Factor ($\alpha$) | 90% | 80% | 70% | 60% | 50% | 0% | Train Signals |
|---|---|---|---|---|---|---|---|
| Herding Welling (2009) | 29.17±0.23 | 41.26±0.43 | 48.71±0.23 | 54.65 ± 0.07 | 58.92 ± 0.19 | 69.52±0.45 | ✗ |
| k-Center Greedy Sener & Savarese (2017) | 48.11±0.29 | 59.06±0.22 | 62.91±0.22 | 64.93 ± 0.22 | 66.04 ± 0.05 | 69.52±0.45 | ✗ |
| Forgetting Toneva et al. (2018) | **55.31**±**0.07** | 60.36±0.12 | 62.45±0.11 | 63.97 ± 0.01 | 65.06 ± 0.02 | 69.52±0.45 | ✓ |
| CALMargatina et al. (2021) | 46.08±0.10 | 53.71±0.19 | 58.11±0.13 | 61.17±0.06 | 63.67 ± 0.28 | 69.52±0.45 | ✓ |
| Craig Mirzasoleiman et al. (2020) | 51.39±0.13 | 59.33±0.22 | 62.72±0.13 | 64.96±0.00 | **66.29 ±0.00** | 69.52±0.45 | ✓ |
| GradMatch Killamsetty et al. (2021a) | 47.57±0.32 | 56.29±0.31 | 60.62±0.28 | 64.40±0.33 | 65.02 ± 0.50 | 69.52±0.45 | ✓ |
| Glister Killamsetty et al. (2021b) | 47.02±0.29 | 55.93±0.17 | 60.38±0.17 | 62.86±0.07 | 65.07±0.08 | 69.52±0.45 | ✓ |
| Facility Location | 52.49±0.19 | 60.06±0.11 | 63.05 ± 0.06 | 65.24 ± 0.04 | 66.05 ± 0.07 | 69.52±0.45 | ✗ |
| **SubZeroCore (ours)** | 54.01±0.14 | **60.78 ±0.05** | **63.35 ±0.11** | **65.32 ±0.04** | 66.14 ±0.07 | 69.52±0.45 | ✗ |

**Main Results.** In Table 2, we show how Sub-ZeroCore compares against existing coreset selection methods on CIFAR-10 under various pruning ratios (from 10% up to 99.9%). Notably, our approach closely matches all baselines for lower pruning rates (70% and below), or consistently outperforms for pruning ratios above 70%, especially for ultra-scarce settings. More details on complexity and additional cross-architecture evaluations (VGG-16 Simonyan & Zisserman (2014), InceptionNetV3 Szegedy et al. (2016), WRN-16-8 Zagoruyko & Komodakis (2016), and ResNet-50 He et al. (2016)) can found in the appendix. Moreover, we achieve all results while being notably faster due to our training-free setup, as shown in Figure 4.

**Comparison to InfoMax.** We also compare SubZeroCore to InfoMax Tan et al. (2025), which extends D2Pruning Maharana et al. (2024) by combining difficulty-based scoring with intermediate convolutional features in an information-maximization objective. Because InfoMax adopts a slightly different post-pruning training protocol than our DeepCore setup, their reported numbers are not directly interchangeable. However, reporting them side by side gives a clearer picture of how geometric, training-free selection compares to a training-dependent alternative. As shown in Table 3, SubZeroCore achieves better performance across all pruning levels, with pronounced gains at extreme pruning ratios (e.g., $\alpha \geq 0.95$), while requiring none of the training-time signals or message-passing used by InfoMax.

**Robustness.** To assess the stability of our coreset selection method under label noise or malicious relabeling, we follow a poisoning protocol similar to that in *Zhang et al.* Zhang et al. (2021). Specifically, we randomly relabel 10% of CIFAR-10 training examples to incorrect classes, thereby introducing a form of data poisoning. We then run each coreset selection method on this poisoned dataset, subsampling different fractions. The relative accuracy change (compared to no poisoning) is shown in Figure 5. We observe that our method SubZeroCore demonstrates profound robustness among all baselines, effectively mitigating the detrimental effects of relabeling noise (i.e., mislabeled data). Notably, its performance remains superior to the standard facility location method. In fact, by incorporating the density-weighted mechanism, our method downweights outlier samples (where mislabeled or corrupted data often lie), yielding a stable coreset even under harsh poisoning scenarios. Such improvements highlight that the density weighting scheme is not only beneficial for standard data selection but also enhances resilience to adversarial or noisy training conditions. Additional evaluations with random relabeling of 20% and 30% can be found in the appendix, where the relative ordering of methods remains largely unchanged.

## 4.2 IMAGENET-1K RESULTS

**Setup.** For our ImageNet-1K experiments, we train three ResNet-18 models on the selected coresets using a batch size of 256 for 200 epochs. Training images are randomly cropped and resized to $224 \times 224$, and horizontal flipping is applied with a probability of 50%. All other experimental settings and training hyperparameters are identical to those used in our CIFAR-10 experiments.

**Main Results.** As shown in Table 4, SubZeroCore consistently ranks among the top-performing methods across all pruning levels, outperforming nearly all training-based approaches. In particular, it matches or slightly exceeds Forgetting at higher pruning ratios and outperforms Craig, GradMatch, and CAL most of the time. Notably, SubZeroCore achieves this performance without any training.

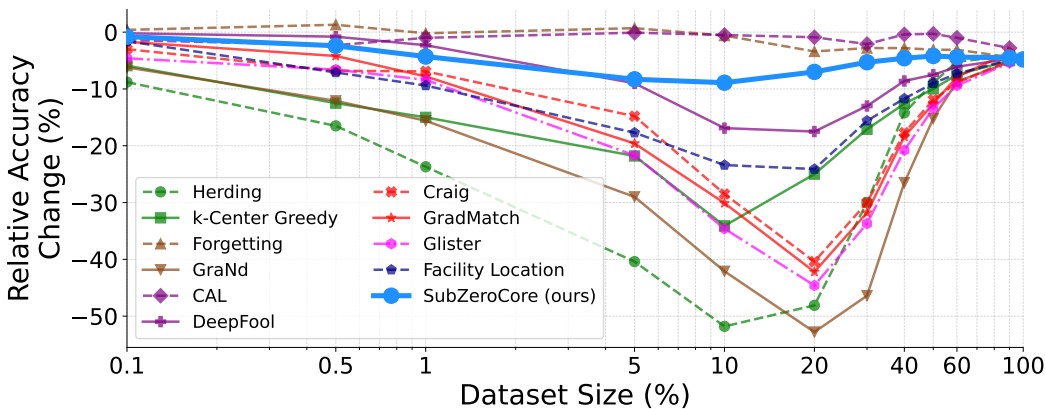

Figure 5: Relative robustness of coreset selection methods on CIFAR-10 with 10% corrupted labels. SubZeroCore demonstrates strong robustness (among top-3 methods with CAL and Forgetting), even outperforming facility location, the method it builds upon.

### 4.3 IMPACT OF TARGET COVERAGE

Recall that SubZeroCore has only one hyperparameter, namely the desired coverage level $\gamma \in (0,1)$ for Equation 9. We conduct an ablation study (see Figure 6) by varying $\gamma$ and then measuring the final test accuracy under different pruning ratios. We observe that, for moderate or low pruning rates, SubZeroCore remains relatively insensitive to the exact choice of $\gamma$. However, at high pruning rates, different $\gamma$-values lead to significant gaps in final accuracy. Through this exploration, we find that a target coverage of $\gamma \approx 0.60$ offers the best trade-off between robust performance and insensitivity to pruning levels. Consequently, we adopted $\gamma = 0.60$ in our reported CIFAR-10 and ImageNet-1K experiments.

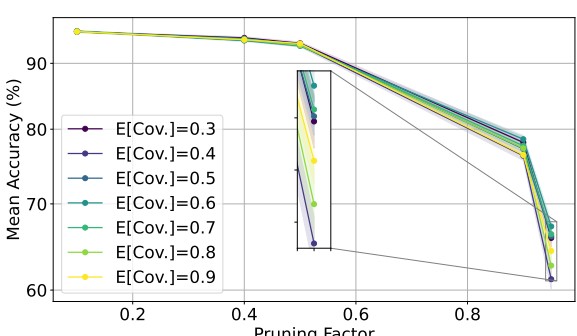

Figure 6: Expected coverage ($\gamma$) ablation on CIFAR-10. While for lower pruning ratios, the setting of $\gamma$ does not have a notable impact, it significantly influences the outcome for higher pruning ratios. We identify a target coverage of 0.6 as the best option.

## 5 LIMITATIONS

SubZeroCore may yield less meaningful estimates in the regime where $|\mathcal{T}|$ is small, although our coverage derivation in Equation 9 cleanly holds for sufficiently large datasets. Mathematically, the closed-form expression hinges on selecting $|\mathcal{S}|$ subsets from a larger pool $|\mathcal{T}|$. When $|\mathcal{T}|$ is only marginally bigger than $|\mathcal{S}|$, the binomial coefficients $\binom{|\mathcal{T}|-K}{|\mathcal{S}|}$ and $\binom{|\mathcal{T}|}{|\mathcal{S}|}$ can be extremely sensitive to small changes in $|\mathcal{S}|$ or $K$. Consequently, small-sample effects can inflate (or deflate) the predicted coverage in ways that do not generalize outside the combinatorial assumptions underlying the derivation. Thus, if the dataset itself is tiny (e.g., tens or hundreds of samples), then the notion of "expected coverage" over all possible subsets becomes so discretized that it no longer provides a stable yardstick for coverage-driven coreset selection. We recommend a direct check of coverage in such low-data scenarios (though it remains questionable whether coreset selection is even necessary in extremely small datasets), rather than relying on the asymptotic-style expression in Equation 9.

While our experiments focus on image classification with moderate-scale architectures, SubZeroCore is not tied to vision-specific inductive biases. The method only requires a fixed embedding space and does not rely on model-dependent training signals. In principle, this makes the approach compatible with other modalities (e.g., text or multimodal data) by operating on embeddings from large pretrained

encoders (e.g., self-supervised vision models or large language models). Exploring these broader settings is an interesting direction for future work, and we include this note here to clarify that SubZeroCore's formulation is inherently domain-agnostic, even though our empirical evaluation is constrained to standard vision benchmarks.

## 6  RELATED WORK

Coreset selection has been explored from multiple angles. On the training-based front, various importance-scoring heuristics like the forgetting score Toneva et al. (2018), AUM Pleiss et al. (2020), and EL2N Paul et al. (2021) estimate how much a training example influences model parameters or loss dynamics, then keep only those deemed most essential. Other methods like GraNd Paul et al. (2021) or GradMatch Killamsetty et al. (2021a) exploit the gradients during training, while DeepFool Ducoffe & Precioso (2018) or CAL Margatina et al. (2021) leverage an approximation of the decision boundary during training. However, computing these metrics usually demands full or partial training rounds and can be computationally heavy. Regarding training-free methods, k-means clustering or greedy k-center have been proposed to directly achieve good coverage in feature space Sener & Savarese (2017); Sorscher et al. (2022), but usually underperform if the embedded feature space is not trained on the full dataset like in our experiments. Also, their sole focus is pure coverage, making it highly effective at covering the entire data space but also sensitive to outliers, as it will prioritize isolated points to reduce the worst-case distance.

Beyond coreset selection specifically, data subsets or proxy selection also appears in active learning, where approaches like BADGE Ash et al. (2019) or BatchBALD Kirsch et al. (2019) repeatedly query diverse, high-uncertainty examples to improve a model at each round. Although active learning shares the goal of sampling efficiently, it typically relies on sequential label querying and repeated model updates, which differ from our training-free, model-agnostic setting. Another relevant line of research pertains to coreset constructions for *classical clustering* problems (e.g., k-means), where theoretical guarantees can be derived through importance sampling or similar randomization strategies Feldman (2020); Cohen-Addad et al. (2025); Bahmani et al. (2012); Caron et al. (2018). These techniques, however, leverage the geometry of clustering objectives rather than classification or representation-learning signals, making them less adaptable to broad deep-learning tasks.

## 7  CONCLUSION & FUTURE WORK

In this paper, we introduced SubZeroCore, a novel coreset selection method that elegantly unifies density and coverage into a single submodular optimization objective without requiring any training signals. Unlike existing training-based methods, SubZeroCore operates sufficiently in a purely geometric-based setting and significantly reduces computational overhead. Moreover, we reduced the number of hyperparameters for the corset selection to one, whereas existing methods rely on good model-specific choices. Our theoretical analysis, supported by extensive experiments on CIFAR-10 and ImageNet-1K, demonstrates that SubZeroCore not only maintains competitive accuracy at lower pruning rates but also outperforms state-of-the-art results at high pruning rates. Moreover, we have shown that our density-based weighting scheme naturally provides robustness against label noise, making it suitable for real-world scenarios with potentially corrupted or noisy data.

In conclusion, SubZeroCore presents a meaningful step forward in making large-scale coreset selection more resource-efficient and environmentally sustainable. Future work includes extending the framework to dynamic data streams, further broadening its applicability. Moreover, one could introduce a additional power on the weights to explicitly control between density and coverage.

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
