# SUPPLEMENTARY MATERIAL FOR SUBZEROCORE: A SUBMODULAR APPROACH WITH ZERO TRAINING FOR CORESET SELECTION

## A    CONNECTING SUBMODULARITY TO GENERALIZATION

In this section, we elaborate on how submodular optimization, specifically through maximizing the facility location function, is inherently connected to generalization performance.

Intuitively, the submodular facility location function encourages the selection of representative and diverse subsets by maximizing the similarity coverage of the entire dataset. Such coverage implies that selected points efficiently span the data manifold, reducing redundancy and increasing representativeness. To formalize this intuition, consider the concept of empirical Rademacher complexity, which quantifies the capacity of a function class to fit random labels and thus directly relates to generalization performance:

**Lemma 1** (Informal connection to generalization). *Given a data distribution $P$, and a submodular coverage function $f(\mathcal{S})$ optimized by greedy submodular maximization to achieve at least a $(1 - 1/e)$ approximation, the resulting coreset $\mathcal{S}_{greedy}$ provides a reduced empirical Rademacher complexity compared to randomly selected subsets of the same size. Consequently, this implies improved expected generalization bounds. Formally,*

$$\mathbb{E}_{\mathbf{x}, y \sim P}[\mathcal{L}(\mathbf{x}, y; \mathcal{S}_{greedy})] \leq \mathbb{E}_{\mathbf{x}, y \sim P}[\mathcal{L}(\mathbf{x}, y; \mathcal{S}_{random})] - \Delta, \tag{1}$$

*where $\Delta \geq 0$ quantifies the expected gain in generalization due to better coverage.*

While a rigorous theoretical derivation of $\Delta$ depends on specific assumptions about the data manifold and similarity measures, the conceptual linkage established here supports the observed empirical improvements in generalization for SubZeroCore.

## B    DENSITY-WEIGHTED SUBMODULARITY AND ROBUSTNESS

This section discusses the theoretical reasoning behind why incorporating density weighting within the submodular selection improves robustness to noise and corrupted labels.

The key theoretical intuition is that emphasizing density inherently focuses the coreset selection on regions of the data manifold with greater local consensus. Points residing in densely populated regions are more likely to reflect stable underlying decision boundaries, making them robust to label perturbations and noise.

**Lemma 2** (Informal connection to robustness). *Given a density-weighted submodular function defined as*

$$f_{SubZeroCore}(\mathcal{S}) = \sum_{i \in \mathcal{T}} \max_{j \in \mathcal{S}} \big( s_j \cdot sim(\mathbf{x}_i, \mathbf{x}_j) \big), \tag{2}$$

*where $s_j$ are density-based weights favoring dense regions, the resulting selected coreset exhibits increased robustness to mislabeled data compared to subsets selected without density weighting. Specifically, the probability that an outlier or noisy label significantly influences the subset decreases substantially:*

$$P(\mathbf{x}_{outlier} \in \mathcal{S}_{greedy}) \leq P(\mathbf{x}_{outlier} \in \mathcal{S}_{non\text{-}density}), \tag{3}$$

*where equality only holds in the trivial case of uniformly dense data.*

This theoretical viewpoint aligns well with empirical observations in our experiments, providing a conceptual rationale for the robustness benefits observed with SubZeroCore.

## C    SUBZEROCORE ALGORITHM

---

**Algorithm 1** SubZeroCore Selection Procedure

---

1: **Input:** Dataset $\mathcal{T} = \{(\mathbf{x}_i, y_i)\}_{i=1}^N$, pruning ratio $\alpha \in (0,1)$, coverage target $\gamma \in (0,1)$, similarity measure $\text{sim}(\cdot, \cdot)$
2: **Output:** Coreset $\mathcal{S}$, with $|\mathcal{S}| = (1-\alpha)|\mathcal{T}|$
3: *// Compute subset size:*
4: $s \leftarrow (1-\alpha) \cdot |\mathcal{T}|$
5: *// Determine optimal $K$ via numerical inversion (given coverage $\gamma$):*
6: $K \leftarrow \min\left\{ K \in \mathbb{N} \;\middle|\; 1 - \gamma \leq \prod_{k=0}^{K} \frac{(|\mathcal{T}|-s-k)}{|\mathcal{T}|-k} \right\}$
7: *// For each $\mathbf{x}_i \in \mathcal{T}$, compute radius:*
8: $r_i \leftarrow \text{NND}_K(\mathbf{x}_i)$
9: *// Compute empirical mean and standard deviation of radii:*
10: $\mu \leftarrow \frac{1}{|\mathcal{T}|} \sum_{i=1}^{|\mathcal{T}|} r_i, \quad \sigma \leftarrow \sqrt{\frac{1}{|\mathcal{T}|} \sum_{i=1}^{|\mathcal{T}|} (r_i - \mu)^2}$
11: *// For each $\mathbf{x}_i \in \mathcal{T}$, compute density-based weights:*
12: $s_i \leftarrow \exp\left( -\frac{(r_i - \mu)^2}{2\sigma^2} \right)$
13:
14: *// Greedily select the coreset by maximizing the weighted facility location:*
15: *// Initialize*
16: $\mathcal{S} \leftarrow \emptyset$
17: **while** $|\mathcal{S}| < s$ **do**
18:     *// Select next element according to:*
19:     $\mathbf{x}^* \leftarrow \arg\max_{\mathbf{x}_j \in \mathcal{T} \setminus \mathcal{S}} \sum_{\mathbf{x}_i \in \mathcal{T}} \max_{\mathbf{x}_k \in \mathcal{S} \cup \{\mathbf{x}_j\}} s_k \cdot \text{sim}(\mathbf{x}_i, \mathbf{x}_k)$
20:     *// Update:*
21:     $\mathcal{S} \leftarrow \mathcal{S} \cup \{\mathbf{x}^*\}$
22: **end while**
23: **return** $\mathcal{S}$

---

## D    NUMERICAL INVERSION OF COVERAGE

To efficiently determine the optimal neighborhood size $K$ given a desired coverage level $\gamma$, we implemented a numerical inversion. Basically, it .... Below we provide the corresponding Python implementation directly extracted from our codebase:

```python
def find_k_for_coverage(M, N, c):
    k = 0
    coverage = 0.0
    numerator = 1.0
    denominator = 1.0
    while coverage < c and k < N - M - 1:
        k += 1
        numerator *= (N - M - k)
        denominator *= (N - k)
        coverage = 1.0 - numerator / denominator
    return k
```

# E  COMPLEXITY ANALYSIS OF SUBZEROCORE

We provide a comprehensive runtime and space complexity analysis for the SubZeroCore algorithm. Below, let:

- $N$ denote the total number of samples in the dataset,

- $C$ denote the number of classes,

- $N_C \approx \frac{N}{C}$ denote the average number of samples per class,

- $d$ denote the dimensionality of the embeddings (e.g., 2048 for Inception v3),

- $M$ denote the coreset size per class (target subset size).

## E.1  RUNTIME COMPLEXITY

SubZeroCore has four main computational steps:

**1. Embedding Extraction.**  Each embedding is computed via a single forward pass through a pretrained model, resulting in complexity:

$$\mathcal{O}(N \cdot d).$$

**2. Numerical Inversion (Finding $K$ for Coverage).**  The numerical inversion (implemented by a simple linear search in practice) has negligible complexity as $K \ll N_C$:

$$\mathcal{O}(K).$$

**3. Computing Radii and Density Scores.**  This involves pairwise distance calculations within each class:

$$\mathcal{O}(N_C^2 \cdot d).$$

Computing density scores (mean and standard deviation) adds only a linear term, thus the complexity is dominated by pairwise distance calculation.

**4. Greedy Facility Location Selection.**  The greedy selection iterates $M$ times, with each iteration recomputing marginal gains for $N_C$ candidates:

$$\mathcal{O}(M \cdot N_C^2).$$

Summarizing all the steps above, the runtime complexity per class is:

$$\mathcal{O}(N_C^2 \cdot d + M \cdot N_C^2) \approx \mathcal{O}(N_C^2 \cdot d),$$

as typically, $M \ll d$. Thus, global complexity (over all classes) is approximately:

$$\mathcal{O}\left(\frac{N^2 \cdot d}{C}\right).$$

## E.2  SPACE COMPLEXITY

SubZeroCore requires storing embeddings and pairwise similarity matrices:

- Embedding storage: $\mathcal{O}(N \cdot d)$,

- Pairwise similarity matrix storage per class: $\mathcal{O}(N_C^2)$.

Thus, the overall memory complexity per class is dominated by the similarity matrix:

$$\mathcal{O}(N_C^2).$$

E.3   WHEN TRAINING-SIGNAL-BASED METHODS CAN HAVE LOWER COMPLEXITY

While SubZeroCore benefits significantly from not requiring any model training, there exist specific scenarios where training-signal-based coreset selection methods may exhibit lower computational complexity:

- **Pre-trained Models or Available Gradients:** If the embedding model or gradient signals are already computed as part of a previous training phase or available from prior runs, methods utilizing these signals (e.g., gradient-based selection or forgetting-based methods) can leverage existing computations to significantly reduce incremental complexity. In these cases, complexity reduces to near-linear in the dataset size.
- **Partial or Incremental Training:** Methods that rely on incremental updates of model parameters, rather than full model retraining, can substantially reduce computational overhead compared to the quadratic complexity involved in calculating pairwise distances or similarities used in SubZeroCore.
- **Small Model Architectures:** For simpler or highly optimized models where obtaining training signals (such as gradients or loss values) can be efficiently computed, training-based methods may become less computationally demanding compared to the quadratic scaling of SubZeroCore.

Therefore, SubZeroCore is generally preferable in large-scale, training-free scenarios or settings where training signals are expensive or unavailable. In contrast, training-signal-based methods could become favorable under conditions outlined above.

## F   CONSIDERATIONS FOR EMBEDDING NETWORKS IN SPECIALIZED DOMAINS

While SubZeroCore fundamentally operates as a training-free coreset selection method, its effectiveness hinges on the quality and suitability of the underlying embedding representation. Throughout our experiments on standard datasets such as CIFAR-10 and ImageNet-1K, we observed negligible performance differences across widely used pre-trained feature extractors like ResNet-18, Inception-NetV3, or VGG. This consistency suggests that SubZeroCore is mainly agnostic to these popular embedding models.

However, for highly specialized datasets or unique domain-specific applications (i.e., medical imaging, remote sensing, or multi-modal data), the choice of embedding network may significantly influence the quality of the selected coreset. Such specialized datasets often present distinct characteristics, such as unique feature distributions, intricate semantic relationships, or uncommon modalities that generic pre-trained embeddings may not adequately capture. In these cases, training a specialized embedding network tailored explicitly for the target domain or task could become necessary to achieve optimal coreset selection performance.

## G   HARDWARE & SOFTWARE DETAILS

All experiments were conducted on a workstation equipped with an NVIDIA RTX A6000 GPU (48GB VRAM) using PyTorch 1.10.1 with torchvision 0.11.2 for DeepCore.

## H   DATASETS

We conduct experiments on two widely adopted image classification datasets: CIFAR-10 and ImageNet-1K. Both datasets have the following properties.

- **CIFAR-10** consists of 50,000 training and 10,000 test images across 10 classes. Each image has a resolution of $32 \times 32$ pixels. The dataset provides a balanced, compact benchmark for testing coreset selection performance under high pruning ratios.
- **ImageNet-1K** is a large-scale dataset with 1,281,167 training and 50,000 validation images from 1,000 object categories. Following standard coreset evaluation practices, we resize all images to $224 \times 224$ resolution for training and testing unless otherwise noted.

## I    ADDITIONAL CROSS-ARCHITECTURE EVALUATIONS

Table 1: Coreset selection performances on CIFAR-10 with five randomly initialized VGG-16 models. The best results are marked in bold.

| Fraction $(1-\alpha)$ | 0.1% | 0.5% | 1% | 5% | 10% | 20% | 30% | 40% | 50% | 60% | 90% | 100% |
|---|---|---|---|---|---|---|---|---|---|---|---|---|
| Herding | 11.6±1.2 | 12.7±0.6 | 16.0±3.9 | 54.1±3.1 | 66.0±3.5 | 71.5±1.5 | 78.3±1.0 | 80.9±0.5 | 85.7±1.5 | 88.2±0.7 | 93.2±0.3 | 94.3±0.0 |
| k-Center Greedy | 12.3±1.0 | 14.5±2.9 | 14.7±1.3 | 38.7±12.0 | 75.6±1.3 | 85.5±0.3 | 89.0±0.4 | 91.0±0.2 | 91.8±0.3 | 92.7±0.2 | 94.1±0.1 | 94.3±0.0 |
| Forgetting | 12.5±0.8 | 13.8±2.8 | **25.6±4.3** | 55.9±1.3 | 75.2±1.4 | 86.2±0.2 | 89.0±0.2 | 91.3±0.2 | 91.9±0.2 | 92.8±0.0 | 94.1±0.2 | 94.3±0.0 |
| GraNd | 11.7±0.6 | 12.5±0.0 | 16.6±3.0 | 28.1±0.0 | 42.4±0.9 | 73.4±0.7 | 86.5±0.7 | 91.2±0.0 | 93.1±0.4 | **93.6±0.2** | 94.1±0.2 | 94.3±0.0 |
| CAL | 12.9±2.0 | 15.8±4.1 | 19.2±4.5 | 63.0±1.4 | 72.2±0.2 | 78.1±0.9 | 82.6±0.5 | 85.0±0.2 | 87.6±0.7 | 89.6±0.4 | 93.0±0.2 | 94.3±0.0 |
| DeepFool | 11.6±0.7 | 14.6±1.3 | 15.6±4.1 | 47.3±6.8 | 71.7±1.1 | 84.2±0.4 | **90.0±0.2** | **91.6±0.1** | 92.4±0.2 | 93.2±0.2 | 94.1±0.1 | 94.3±0.0 |
| Craig | 13.3±1.4 | 14.2±1.4 | 13.6±0.0 | 46.6±0.0 | 67.3±3.1 | 82.4±0.6 | 88.8±0.3 | 91.4±0.0 | 92.7±0.2 | 93.4±0.1 | **94.2±0.1** | 94.3±0.0 |
| GradMatch | 13.4±2.0 | 13.0±2.3 | 13.0±0.0 | 44.6±3.7 | 53.9±1.4 | 77.9±0.4 | 87.2±0.3 | 90.2±0.6 | **92.6±0.0** | 93.4±0.0 | **94.2±0.1** | 94.3±0.0 |
| Glister | **15.7±0.0** | 13.9±3.0 | 18.9±5.0 | 44.0±4.0 | 53.5±0.9 | 78.2±2.2 | 86.4±0.1 | 89.7±0.0 | 92.5±0.2 | 93.3±0.2 | **94.2±0.1** | 94.3±0.0 |
| Facility Location | 12.6±1.3 | 16.8±2.5 | 15.3±2.5 | 48.0±6.9 | 73.4±0.7 | 85.4±0.3 | 89.5±0.3 | 90.6±0.3 | 92.5±0.2 | 93.2±0.1 | 94.1±0.1 | 94.3±0.0 |
| **SubZeroCore (ours)** | 12.8±0.6 | **18.0±0.0** | 17.7±3.2 | **65.7±1.7** | **78.1±0.4** | **86.9±0.2** | 88.8±0.3 | 90.7±0.1 | 91.9±0.1 | 92.2±0.0 | 93.7±0.1 | 94.3±0.0 |

Table 2: Coreset selection performances on CIFAR-10 with five randomly initialized ResNet-50 models. The best results are marked in bold.

| Fraction $(1-\alpha)$ | 0.1% | 0.5% | 1% | 5% | 10% | 20% | 30% | 40% | 50% | 60% | 90% | 100% |
|---|---|---|---|---|---|---|---|---|---|---|---|---|
| Herding | 16.2±2.4 | 23.0±2.4 | 29.5±2.8 | 46.3±2.8 | 58.5±5.2 | 73.1±2.1 | 80.0±0.9 | 84.1±0.5 | 86.4±0.9 | 89.6±0.9 | 94.6±0.3 | 95.4±0.3 |
| k-Center Greedy | 15.4±1.8 | 18.8±3.1 | 25.0±4.6 | 53.1±1.9 | 68.8±2.1 | 86.6±0.4 | 90.2±0.6 | 91.7±0.2 | 93.4±0.3 | 94.2±0.3 | 95.3±0.3 | 95.4±0.3 |
| Forgetting | 17.0±2.2 | 26.9±2.7 | 34.0±1.0 | 49.5±2.9 | 67.5±2.6 | 86.2±0.7 | 89.9±0.4 | 92.0±0.3 | 92.8±0.2 | 93.6±0.1 | 95.2±0.1 | 95.4±0.3 |
| GraNd | 15.1±1.7 | 17.2±1.4 | 19.7±1.2 | 23.6±1.6 | 34.9±2.4 | 70.3±5.0 | 85.5±1.4 | 92.0±0.3 | **94.2±0.4** | **94.9±0.4** | **95.4±0.2** | 95.4±0.3 |
| CAL | 19.9±1.3 | 27.5±0.6 | 35.9±1.2 | 56.4±1.4 | 66.2±1.1 | 77.6±0.8 | 83.0±0.4 | 85.5±0.5 | 88.2±0.5 | 90.5±0.6 | 94.2±0.1 | 95.4±0.3 |
| DeepFool | 14.4±2.0 | 19.8±1.4 | 26.4±1.1 | 42.7±2.9 | 65.7±2.8 | 86.3±0.9 | **91.1±0.5** | **92.9±0.3** | 93.9±0.2 | 94.5±0.2 | 95.3±0.2 | 95.4±0.3 |
| Craig | 17.0±1.1 | 21.5±1.4 | 28.1±2.6 | 42.0±1.5 | 55.1±2.7 | 80.8±2.8 | 89.9±0.7 | 92.5±0.7 | 93.5±0.2 | 94.7±0.1 | **95.4±0.2** | 95.4±0.3 |
| GradMatch | 14.3±2.8 | 21.7±0.9 | 28.2±0.9 | 37.4±2.6 | 50.5±3.5 | 74.0±2.3 | 87.6±1.1 | 92.3±0.4 | 93.8±0.3 | 94.7±0.5 | 95.2±0.2 | 95.4±0.3 |
| Glister | 14.9±0.9 | 20.4±1.5 | 24.2±3.3 | 37.0±1.9 | 50.1±3.0 | 79.6±2.5 | 88.6±0.8 | 92.0±0.3 | 93.5±0.2 | 94.5±0.3 | 95.2±0.2 | 95.4±0.3 |
| Facility Location | 16.6±1.4 | 20.1±4.0 | 32.9±2.0 | 55.9±2.4 | 71.1±3.6 | 86.7±1.0 | 90.8±0.4 | 92.7±0.2 | 93.7±0.5 | 94.4±0.3 | 95.1±0.3 | 95.4±0.3 |
| **SubZeroCore (ours)** | **20.1±0.5** | **27.8±2.0** | **36.5±0.7** | **57.6±3.3** | **74.6±0.9** | **86.8±0.7** | 90.0±0.2 | 91.4±0.3 | 92.8±0.3 | 93.3±0.3 | 95.3±0.1 | 95.4±0.3 |

Table 3: Coreset selection performances on CIFAR-10 with five randomly initialized InceptionNetV3 models. The best results are marked in bold.

| Fraction $(1-\alpha)$ | 0.1% | 0.5% | 1% | 5% | 10% | 20% | 30% | 40% | 50% | 60% | 90% | 100% |
|---|---|---|---|---|---|---|---|---|---|---|---|---|
| Herding | 14.6±0.7 | 21.8±1.7 | 28.5±1.4 | 42.9±4.0 | 61.1±2.2 | 74.0±1.3 | 80.5±1.0 | 84.7±0.4 | 87.7±0.9 | 90.3±0.8 | 94.8±0.2 | 95.6±0.1 |
| k-Center Greedy | 15.1±2.0 | 22.1±3.0 | 26.6±1.4 | 51.2±2.3 | 72.9±1.7 | 85.8±0.4 | 89.8±0.4 | 92.2±0.3 | 93.6±0.3 | 94.4±0.4 | 95.4±0.2 | 95.6±0.1 |
| Forgetting | 18.5±0.6 | 28.5±1.0 | 32.2±0.9 | 50.9±1.5 | 69.3±1.3 | **85.4±0.9** | **90.4±0.4** | 92.4±0.1 | 93.4±0.2 | 94.3±0.3 | 95.4±0.2 | 95.6±0.1 |
| GraNd | 14.8±2.3 | 17.4±1.2 | 20.1±0.8 | 27.1±1.1 | 38.1±1.4 | 70.0±2.0 | 86.8±0.6 | **92.9±0.4** | **94.4±0.3** | **95.0±0.3** | 95.5±0.1 | 95.6±0.1 |
| CAL | 17.9±1.2 | 31.6±1.5 | 36.3±1.1 | 60.2±2.0 | 70.6±0.6 | 78.7±0.7 | 83.1±0.5 | 86.3±0.8 | 88.9±0.4 | 90.3±0.3 | 94.3±0.1 | 95.6±0.1 |
| DeepFool | 14.5±1.6 | 22.3±1.1 | 26.8±1.6 | 47.7±3.0 | 70.2±2.1 | 83.6±1.2 | 90.2±0.4 | 92.8±0.2 | 94.0±0.2 | 94.6±0.3 | **95.6±0.2** | 95.6±0.1 |
| Craig | 16.5±2.5 | 24.7±1.3 | 27.8±2.0 | 44.5±1.8 | 58.8±2.5 | 79.5±1.4 | 88.0±1.1 | 92.3±0.2 | 94.0±0.2 | **95.0±0.2** | **95.6±0.1** | 95.6±0.1 |
| GradMatch | 16.5±1.2 | 24.2±0.8 | 29.1±1.0 | 40.1±2.7 | 53.1±2.2 | 76.9±2.6 | 87.1±0.3 | 91.8±0.7 | 94.2±0.5 | 94.9±0.3 | 95.4±0.2 | 95.6±0.1 |
| Glister | 14.8±1.5 | 22.9±1.8 | 29.3±1.8 | 41.3±1.8 | 52.4±3.1 | 77.5±1.7 | 87.8±0.7 | 91.8±0.4 | 94.0±0.1 | 94.9±0.1 | **95.6±0.1** | 95.6±0.1 |
| Facility Location | **18.7±2.7** | 26.4±2.5 | 34.4±1.8 | 59.7±2.5 | 73.1±2.1 | 85.2±0.9 | 89.8±0.1 | 92.8±0.2 | 93.9±0.2 | 94.8±0.2 | 95.5±0.2 | 95.6±0.1 |
| **SubZeroCore (ours)** | **18.7±1.6** | **30.7±0.8** | **36.7±1.5** | **63.3±0.4** | **76.2±1.2** | **85.4±0.4** | 89.4±0.5 | 91.7±0.2 | 92.7±0.1 | 93.7±0.1 | 95.3±0.1 | 95.6±0.1 |

## J    SUBZEROCORE IS SUBMODULAR

We referred to *Berczi et al.* and omitted the proof in the main paper. For completeness, the proof can be done like the following.

**Corollary 1.** *The SubZeroCore function $f_{SubZeroCore}$ is submodular.*

*Proof.* Let $A \subseteq B \subseteq \mathcal{T}$ and pick any $k \in \mathcal{T} \setminus B$. For each $i \in \mathcal{T}$, define

$$\Delta_i(k \mid A) = \max\big(s_k \cdot \mathrm{sim}(\mathbf{x}_i, \mathbf{x}_k), \max_{j \in A} s_j \cdot \mathrm{sim}(\mathbf{x}_i, \mathbf{x}_j)\big) - \max_{j \in A} s_j \cdot \mathrm{sim}(\mathbf{x}_i, \mathbf{x}_j).$$

Then the total marginal gain is

$$f(A \cup \{k\}) - f(A) = \sum_{i \in \mathcal{T}} \Delta_i(k \mid A), \quad f(B \cup \{k\}) - f(B) = \sum_{i \in \mathcal{T}} \Delta_i(k \mid B).$$

Table 4: Coreset selection performances on CIFAR-10 with five randomly initialized WRN-16-8 models. The best results are marked in bold.

| Fraction $(1-\alpha)$ | 0.1% | 0.5% | 1% | 5% | 10% | 20% | 30% | 40% | 50% | 60% | 90% | 100% |
|---|---|---|---|---|---|---|---|---|---|---|---|---|
| Herding | 23.8±1.4 | 31.3±3.5 | 39.6±2.8 | 60.2±1.2 | 66.9±2.3 | 77.3±1.0 | 81.9±0.7 | 86.4±1.3 | 89.0±0.6 | 92.3±1.0 | 95.2±0.1 | 96.0±0.1 |
| k-Center Greedy | 19.2±0.6 | 27.7±0.6 | 34.6±0.2 | 67.9±0.8 | 81.6±0.5 | 89.4±0.2 | 92.1±0.3 | 93.6±0.2 | 94.3±0.1 | 94.9±0.1 | 95.9±0.1 | 96.0±0.1 |
| Forgetting | 21.5±1.0 | 30.1±0.9 | 36.3±0.6 | 58.5±1.0 | 74.9±0.8 | 88.8±0.5 | **93.3±0.1** | **94.5±0.1** | 95.0±0.1 | 95.4±0.1 | 95.9±0.1 | 96.0±0.1 |
| GraNd | 15.0±1.8 | 19.5±0.7 | 21.4±0.3 | 37.8±1.0 | 58.7±1.0 | 81.8±0.7 | 92.1±0.1 | 94.3±0.2 | **95.3±0.1** | **95.7±0.2** | **96.0±0.2** | 96.0±0.1 |
| CAL | 22.2±2.4 | 37.1±2.5 | 45.5±1.4 | 66.2±0.6 | 74.2±0.9 | 82.7±0.5 | 86.3±0.8 | 89.2±0.6 | 91.0±0.3 | 92.4±0.2 | 95.3±0.1 | 96.0±0.1 |
| DeepFool | 18.9±1.9 | 29.2±1.0 | 35.0±1.8 | 57.4±3.2 | 74.0±1.8 | 87.5±0.4 | 92.0±0.3 | 93.5±0.2 | 94.6±0.1 | 95.2±0.1 | **96.0±0.1** | 96.0±0.1 |
| Craig | 21.9±1.5 | 30.0±0.8 | 38.1±1.4 | 60.1±1.4 | 69.2±0.8 | 86.5±0.4 | 91.4±0.2 | 93.8±0.1 | 94.8±0.1 | 95.4±0.1 | 95.9±0.1 | 96.0±0.1 |
| GradMatch | 20.1±1.8 | 27.8±1.2 | 31.4±2.1 | 54.4±2.6 | 70.5±2.0 | 84.6±0.7 | 90.7±0.5 | 93.3±0.3 | 94.7±0.4 | 95.3±0.1 | **96.0±0.1** | 96.0±0.1 |
| Glister | 20.1±1.2 | 28.0±1.4 | 31.6±0.8 | 49.3±3.3 | 67.9±1.7 | 83.6±0.8 | 90.5±0.9 | 93.6±0.1 | 94.6±0.1 | 95.2±0.1 | **96.0±0.2** | 96.0±0.1 |
| Facility Location | 25.8±1.8 | 37.4±0.3 | 46.8±1.0 | 75.1±0.6 | 82.1±0.5 | 89.0±0.1 | 92.4±0.2 | 93.7±0.1 | 94.7±0.1 | 95.1±0.1 | 95.9±0.1 | 96.0±0.1 |
| **SubZeroCore (ours)** | **26.6±0.6** | **38.1±0.2** | **47.9±0.5** | **75.6±0.4** | **83.1±0.3** | **89.5±0.2** | 91.8±0.1 | 93.2±0.1 | 93.9±0.1 | 94.5±0.1 | 95.9±0.1 | 96.0±0.1 |

Since $\max_{j\in A} s_j \cdot \mathrm{sim}(\mathbf{x}_i, \mathbf{x}_j) \leq \max_{j\in B} s_j \cdot \mathrm{sim}(\mathbf{x}_i, \mathbf{x}_j)$, it follows for each $i$ that $\Delta_i(k \mid A) \geq \Delta_i(k \mid B)$. Summing over $i$ gives

$$f(A \cup \{k\}) - f(A) \ \geq \ f(B \cup \{k\}) - f(B),$$

which is exactly the diminishing-returns condition for submodularity. ∎

## K    ADDITIONAL ROBUSTNESS EVALUATIONS

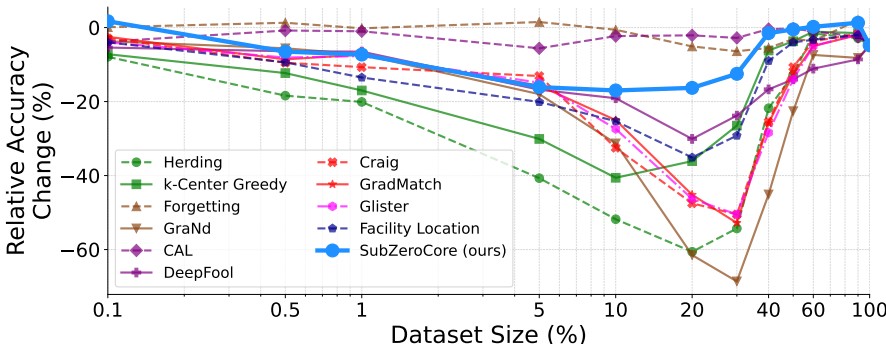

Figure 1: Relative robustness of coreset selection methods on CIFAR-10 with 20% corrupted labels. SubZeroCore demonstrates strong robustness (among top-3 methods with CAL and Forgetting), even outperforming facility location, the method it builds upon.

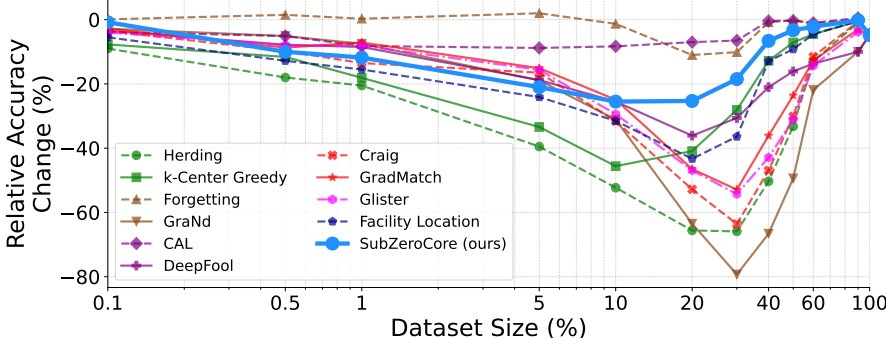

Figure 2: Relative robustness of coreset selection methods on CIFAR-10 with 30% corrupted labels. SubZeroCore demonstrates strong robustness (among top-3 methods with CAL and Forgetting), even outperforming facility location, the method it builds upon.

## L  WHY DENSITY NORMALIZATION REDUCES SENSITIVITY TO NOISE

K-NN based density estimates can be sensitive to noise or outliers. SubZeroCore reduces this sensitivity through its normalized score

$$s_i = \exp\left(-\frac{(r_i - \mu)^2}{2\sigma^2}\right),$$

where $r_i$ is the K-NN radius of sample $i$ and $\mu, \sigma$ are the empirical mean and standard deviation of all radii in the class.

**1. Isolation penalty.** Noisy points often appear as geometric outliers and yield large radii $r_i \gg \mu$. The Gaussian factor assigns them scores near zero, which prevents them from being selected even if they increase coverage.

**2. Suppression of over-dense pockets.** Small local noise clusters produce unusually small radii $r_i \ll \mu$. The same normalization also downweights these values, so tiny clusters of mislabeled samples do not receive high scores.

**3. Stability from concentration of measure.** In typical embedding spaces, most samples lie near the mean radius. For these samples with $r_i \approx \mu$, the score remains close to one and varies smoothly. Consequently, noise-induced fluctuations in radii have limited influence.

**4. Effect on the objective.** The weighted facility-location function

$$f(\mathcal{S}) = \sum_{\mathbf{x} \in \mathcal{T}} \max_{\mathbf{x}_j \in \mathcal{S}} \big(s_j \cdot \text{sim}(\mathbf{x}, \mathbf{x}_j)\big)$$

inherits this robustness. Samples with small $s_j$ contribute little to the objective even if they are isolated or form small dense pockets.

## M  SUBZEROCORE AND ACTIVE LEARNING

Active learning and supervised coreset pruning both select informative samples, but they target different goals. Active learning assumes a large pool of *unlabeled* data and seeks to reduce labeling cost by querying which points to label next. Subset quality is therefore evaluated through model uncertainty or label acquisition efficiency.

Our work follows the supervised coreset setting, where all data are already labeled and the objective is to prune the dataset efficiently. SubZeroCore is fully training-free and uses only geometric properties in embedding space to score samples. Class labels are used solely to run selection class-wise, not for computing importance scores.

Because the two settings optimize for different objectives (i.e., label acquisition vs. dataset reduction), their methods and evaluations are not directly comparable. SubZeroCore is designed specifically for the supervised pruning scenario, and our results should be interpreted within that framework.

## N  RELATION TO OTHER SUBMODULAR OBJECTIVES

Classical monotone submodular functions such as GraphCut, Disparity-Min, and FLMI (Facility Location Mutual Information) provide complementary perspectives on diversity and coverage. GraphCut encourages selecting points that are dissimilar from the remaining dataset, while Disparity-Min focuses on maximizing pairwise spread within the selected subset. FLMI augments the facility-location objective with a mutual-information term that balances representativeness and diversity.

SubZeroCore differs from these objectives in two key ways. First, it retains then facility-location structure but introduces a density-weighted formulation that systematically emphasizes samples in moderately dense regions rather than uniformly prioritizing coverage or diversity. Second, its radius parameter is derived from a closed-form coverage estimate, providing an interpretable and data-dependent way to control the balance between coverage and density, an explicit mechanism not present in GraphCut, Disparity-Min, or FLMI.

While the building blocks used in SubZeroCore, such as $K$-nearest neighbor radii and classical volume estimates, are well established, the contribution lies in how these quantities are combined into a single, training-free coreset objective. The density-weighted facility-location function unifies global coverage and local density, and the closed-form coverage estimator eliminates the need for tuning $K$ or relying on model-dependent signals. This integration yields a lightweight, hyperparameter-efficient alternative to training-based coresets, while preserving the theoretical guarantees of submodular maximization.