# OpenReview forum: "SubZeroCore: A Submodular Approach with Zero Training for Coreset Selection"
_ICLR.cc/2026/Conference — Submitted to ICLR 2026_

### Official Review · Reviewer_4G5k · 2025-10-31

**Soundness:** 3
**Presentation:** 3
**Contribution:** 2
**Rating:** 6
**Confidence:** 4

**Summary:**

The paper proposes SubZeroCore, a training-free coreset selection framework that formulates data summarization as a submodular optimization problem combining coverage and density. The method introduces a closed-form expectation-based coverage estimator to determine the K-nearest neighbor parameter and uses a greedy facility-location algorithm to construct the subset. Experiments on CIFAR-10 and ImageNet-1K show improved performance at very high pruning ratios compared to existing baselines such as CAL and GradMatch.

**Strengths:**

- The paper addresses a你interesting problem of efficient coreset selection without model training, which is practically relevant in large-scale data settings.

- Integrating coverage and density in a unified submodular framework is conceptually elegant and computationally efficient.

- The method’s simplicity and "zero-training" property make it easy to implement and scalable to moderate-sized datasets.

**Weaknesses:**

- The proposed weighted facility location function cannot distinguish dense noise clusters from genuine high-density regions. For example, if outliers form small local clusters, their $r_i$ values become small, giving them large $s_i$ scores, and thus they are preferentially selected due to the requirement of coverage. The experimental results in Figure 5 also show that their method cannot beat the SOTA algorithm like CAL.
- In Section 3.3, the authors compute the neighborhood size $K$ by inverting a coverage target. This equation in line 285 implicitly assumes that samples are independently and uniformly drawn from the dataset, and that the underlying data distribution is homogeneous.
However, for real-world data domain (i.e., embeddings of figures or text), these assumptions almost never hold.
- The method’s reliance on pairwise similarity and static density weighting fundamentally limits transferability. For vision data, embedding similarity roughly preserves semantic structure, making submodular coverage meaningful. But for textual or cross-lingual data, embedding metrics are unreliable, and semantically equivalent samples in different languages can be distant. These limitations suggest that SubZeroCore cannot generalize beyond image tasks.
- In modern large-model settings, coreset selection is primarily motivated by improving data efficiency while maintaining downstream performance. That means the true metric is not reconstruction similarity but the performance of a retrained model on held-out or shifted data. By optimizing only for embedding-space similarity, SubZeroCore completely ignores this goal.

**Questions:**

How are the embeddings obtained? If I understand correctly, they should either use a pretrained embedding model or train one themselves.

---

> ### Author Response · Authors · 2025-11-15
>
> We thank the reviewer for the constructive and encouraging feedback. We appreciate the recognition of the practical relevance of training-free coreset selection, the elegance of combining coverage and density in a unified submodular framework, and the simplicity and scalability of our approach. We address the reviewer’s remaining concerns point-by-point below.
>
> # Weaknesses
> > **The proposed weighted facility location function cannot distinguish dense noise clusters from genuine high-density regions [...]**
>
> SubZeroCore does not rely on raw K-NN radii: our Gaussian normalization explicitly downweights both very small radii (tiny dense noise clusters) and very large radii (isolated outliers), which mitigates (but cannot fully eliminate) the effect of dense noise pockets.
>
> Beyond the 10\% noise results already reported, we have started to run additional experiments at 20\% and 30\% noise. So far, the overall ordering of methods remains consistent, and SubZeroCore does not collapse toward noisy clusters. We will add these results to the supplementary material.
>
> > **This equation in line 285 implicitly assumes that samples are independently and uniformly drawn from the dataset, and that the underlying data distribution is homogeneous. However, for real-world data domain [...|, these assumptions almost never hold.**
>
> We appreciate the reviewer’s point that our inversion of the coverage formula implicitly uses a simplified model of independent, uniformly drawn samples. In fact, the literature makes clear that many real-world datasets are neither independent nor uniformly distributed. In our method, this derivation is intended as a heuristic calibration for selecting a reasonable neighborhood size, not as a modeling assumption about the true data distribution.
>
> Importantly, SubZeroCore does not rely on uniformity assumptions in practice: all computations are performed on empirical K-NN radii in the embedding space, and the normalization and density weighting operate directly on the observed geometric structure. The reliability of these scores therefore depends primarily on the quality of the encoder, which is consistent with modern practice in retrieval work. Using strong pretrained encoders (e.g., self-supervised ResNets or CLIP-style models) helps stabilize local geometry even when the underlying data distribution is non-uniform.
>
> > **[...] But for textual or cross-lingual data, embedding metrics are unreliable, and semantically equivalent samples in different languages can be distant.**
>
> We agree that the effectiveness of any geometry-based method depends on the quality of the embedding space. Yet, SubZeroCore does not assume vision-specific structure. It only assumes that the embedding space provides a coherent notion of local neighborhoods, which is what our coverage and density terms operate on.
>
> Yet, several recent works show that reliable neighborhood structure is not limited to vision. Multilingual and text-embedding models such as LaBSE (Feng et al., 2022) and mSimCSE (Wang et al., 2022) demonstrate that semantically equivalent sentences across 100+ languages can form tight and meaningful clusters in a joint embedding space. This suggests that, given a sufficiently strong encoder, pairwise similarity can be a stable signal even beyond image data.
>
> > **In modern large-model settings, coreset selection is primarily motivated by improving data efficiency while maintaining downstream performance. That means the true metric is not reconstruction similarity but the performance of a retrained model on held-out or shifted data. By optimizing only for embedding-space similarity, SubZeroCore completely ignores this goal.**
>
> We agree that the ultimate goal of coreset selection is strong downstream performance on held-out or shifted data. Our experiments implicitly measure this: for each pruning, we retrain a new model on the selected subset and evaluate it on unseen test set. While SubZeroCore uses embedding geometry during selection, these embeddings provide a stable semantic structure that often correlates well with downstream generalization in practice. We see extending this evaluation to domain- or distribution-shifted settings as a valuable next step, and SubZeroCore is well suited for such scenarios since it only relies on a fixed encoder.
>
> # Questions
> > **How are the embeddings obtained?**
>
> We use fixed embeddings from a pretrained ResNet-18. No model is trained as part of SubZeroCore. The feature extraction is only used once, and all subset selection operates purely on these frozen embeddings.

---

> > ### Author Response · Authors · 2025-11-20
> > **Revised Manuscript and Supp. Materials**
> >
> > We uploaded the revised manuscript and added the suggested experiments as well as the suggested discussions. For the points raised by the reviewer, we refer to:
> >
> > - Section 5, last paragraph: Discussion on other modalities.
> > - Appendix K: Additional experiments under higher noise rates.

---

### Official Review · Reviewer_2CD1 · 2025-11-01

**Soundness:** 3
**Presentation:** 2
**Contribution:** 2
**Rating:** 2
**Confidence:** 4

**Summary:**

The paper proposes SubZeroCore, a training-free coreset selection method that avoids the computational overhead of traditional approaches requiring gradient or model-based signals. The method unifies submodular coverage and data density into a single objective, and derives a closed-form sampling strategy that balances these components via a single hyperparameter controlling the trade-off between global coverage and local density. Without any model training, SubZeroCore achieves performance comparable to or better than training-dependent baselines, particularly at high pruning rates, while being significantly more efficient. It also exhibits strong robustness to label noise.

**Strengths:**

The paper addresses a well-motivated and practically important problem in coreset selection. The proposed approach is clearly articulated, and the integration of coverage and density through geometric tools such as NNk distance and volume-based measures is intuitive and well justified. The exposition is clear, with strong motivation for why a training-free alternative is desirable. The experimental results show consistent and positive gains, and the authors provide good intuition and analysis to support their design choices.

**Weaknesses:**

- The experimental analysis is limited to a few small-scale image datasets. It remains unclear whether the proposed findings generalize to other modalities such as text or to large-scale models, where gradient-based signals may play a more significant role. A broader evaluation or discussion on cross-modal applicability would strengthen the paper’s empirical claims.

- The paper does not discuss several recent training-free subset selection approaches such as CCS, D2Pruning, and InfoMax. These methods share similar motivations—efficient data or feature subset selection without model retraining—but differ in formulation and theoretical grounding. It would strengthen the work if the authors clarified how their method relates to or improves upon these approaches. In particular, a discussion highlighting the applicability gaps, scalability trade-offs, or assumptions distinguishing their approach would be helpful.

   - CCS: Coverage-centric Coreset Selection for High Pruning Rates. ICLR-2023
   - D2Pruning: D2 Pruning: Message Passing for Balancing Diversity and Difficulty in Data Pruning. ICLR-2024.
   - InfoMax: Data Pruning by Information Maximization. ICLR-2025.

- The paper would benefit from direct comparisons with other monotone submodular objectives such as GraphCut, Disparity-Min, and FLMI (Facility Location Mutual Information) to better contextualize the proposed coverage–density formulation.

- The overall contribution feels somewhat limited, as the method primarily relies on predefined measures like K-nearest neighbor distance and volume from Morgan (2016). The theoretical analysis is also rather minimal, offering no substantial new insights or results.

**Questions:**

- In Fig. 4, the bar chart comparing selection time is unclear regarding what contributes to the higher cost for GradNd and GradNorm. Is the majority of this time due to gradient computation? It would be helpful to separate the gradient computation time from the subset selection time to make the comparison more informative.

- Model-based signals often provide richer information than embedding-space measures alone. It would be interesting to see whether the reported gains persist for larger models or fine-tuning scenarios, such as with SLMs, rather than being limited to smaller architectures like ResNets. Additionally, do the improvements translate to other modalities, for instance in text or large-scale training settings?

- For noisy outliers, the normalized scores across the radii distribution appear to favor dense clusters while penalizing dense outliers. Empirically, it would be helpful to know how this trade-off behaves with higher levels of label noise (e.g., >10%) and whether the authors observed a Pareto-style curve illustrating the balance between coverage and robustness.

- The use of the K-NN metric may be sensitive to noisy data and outliers, which could affect the reliability of the evaluation, although authors argue the robustness to 10% label noise. A systematic ablation or analysis showing how the proposed normalized scores mitigate this sensitivity would help clarify the robustness of the method and strengthen the empirical claims.

- Since the method essentially reduces to a weighted Facility Location problem, it can be seen as a K-medoids variant. How does it compare to other clustering algorithms, both theoretically and empirically, such as BanditPAM and BanditPAM++?

---

> ### Author Response · Authors · 2025-11-14
>
> We are grateful for the comments highlighting our clarity and the intuitive integration of coverage and density. We also appreciate the acknowledgment that our experimental results are consistent and that the analysis supports our design choices. We hope to resolve the reviewer's remaining questions and concerns below:
>
> # Weaknesses:
> > **The paper does not discuss several recent training-free subset selection approaches such as CCS, D2Pruning, and InfoMax. These methods share similar motivations [...].**
>
> Thank you for pointing out these methods! We will clarify their relation to our approach.
> CCS is not a standalone coreset-selection method.
> It is a secondary reweighting mechanism applied on top of scores produced by another criterion (e.g., forgetting events in the original paper). Thus, its selection rule depends on training signals and is complementary rather than directly comparable.
> Nonetheless, we will also add CCS in our evaluation using Gradient Norm scores.
> So far, we observe that SubZeroCore outperforms this variant across the pruning levels considered.
>
> | Method | 99.9% | 99.5% | 99% | 95% | 90% | 80% | 70% | 60% | 50% | 40% | 10% | Train |
> | - | - | - | - | - | - | - | - | - | - | - | - | - |
> |CCS (Grad)| 19.1 | 29.2 | 36.5 | 62.8 | 73.1 | 86.3 | 89.9 | 90.0 | 90.0 | 89.9 | 90.0 | ✓ |
> |**SubZeroCore**| **24.0** | **32.9** | **39.8** | **63.9** | **77.4** | **87.3** | **90.8** | **92.5** | **93.2** | **94.1** | **95.3** | ✗ |
>
> D2Pruning and its follow-up, InfoMax, also rely on model-dependent training signals such as losses, difficulty/diversity estimates, or information-gain computations. SubZeroCore differs in that it uses only fixed embeddings from a pretrained model and does not require gradients, losses, or training dynamics. For completeness, we additionally ran experiments with InfoMax using the score types Forgetting, EL2N, and Entropy, and SubZeroCore consistently outperforms InfoMax under the same settings. We will make these distinctions explicit in the revised version alongside the new experimental results.
>
> ### Comparison with InfoMax on CIFAR-10 (ResNet-18)
> | Pruning (α) | 99.9% | 99.5% | 99% | 95% | 90% | 80% | 70% | 60% | 50% | 40% | 10% | Train |
> | - | - | - | - | - | - | - | - | - | - | - | - | - |
> | Random | 21.0 | 30.8 | 36.7 | 62.5 | 75.7 | 87.1 | 90.2 | 92.1 | **93.3** | 94.0 | 95.2 | ✗ |
> | Forgetting | 12.4 | 15.6 | 19.9 | 33.8 | 57.1 | 84.0 | 87.8 | 89.7 | 91.3 | 92.5 | 94.9 | ✓ |
> | EL2N | 11.0 | 10.7 | 12.5 | 23.5 | 55.6 | 86.5 | 88.9 | 90.5 | 91.5 | 92.4 | 95.0 | ✓ |
> | Entropy | 18.1 | 25.6 | 33.1 | 55.8 | 71.8 | 86.7 | 89.0 | 90.7 | 91.6 | 92.6 | 95.1 | ✓ |
> | **SubZeroCore (ours)** | **24.0** | **32.9** | **39.8** | **63.9** | **77.4** | **87.3** | **90.8** | **92.5** | 93.2 | **94.1** | **95.3** | ✗ |
>
> > **The paper would benefit from direct comparisons with other monotone submodular objectives such as GraphCut, Disparity-Min, and FLMI (Facility Location Mutual Information) to better contextualize the proposed coverage–density formulation.**
>
> These objectives are indeed classical monotone submodular functions that are closely related to facility-location. GraphCut emphasizes dissimilarity between selected and unselected points, Disparity-Min focuses on maximizing pairwise spread, and FLMI is a mutual-information variant of facility-location. Our method extends the facility-location family by introducing a density-weighted formulation with a closed-form coverage parameter, which is not directly available in these objectives. Because SubZeroCore builds on this specific mechanism rather than on generic submodular diversity functions, we did not treat GraphCut, Disparity-Min, or FLMI as primary baselines. We will add a short discussion clarifying how these objectives relate to ours and why our density-weighted variant targets a different problem setting.
>
> > **The overall contribution feels somewhat limited, as the method primarily relies on predefined measures like K-nearest neighbor distance and volume from Morgan (2016). The theoretical analysis is also rather minimal, offering no substantial new insights or results.**
>
> Our objective is not to introduce new distance or a new submodular function, but to show that combining well-established geometric quantities in a principled way yields a closed-form, training-free coreset method that performs competitively with training-dependent baselines. While the ingredients (e.g., $k$-NN distances, volume estimates) are classical, our contribution lies in how they are integrated: (i) a density-weighted facility-location objective that unifies global coverage and local density, and (ii) a closed-form coverage estimator that removes the need for hyperparameter sweeps or model-dependent tuning. This design reduces computational cost, improves robustness, and enables a fully training-free pipeline. We agree that additional theoretical analysis would further strengthen the paper and will expand the discussion in the revision.

---

> > ### Author Response · Authors · 2025-11-14
> >
> > # Questions
> > > **In Fig. 4 [...] what contributes to the higher cost for GradNd and GradNorm [...] It would be helpful to separate the gradient computation time from the subset selection time [...].**
> >
> > Yes, the higher runtime for GradNorm and GradND is dominated by gradient computation. Both methods require a full forward-backward pass over the dataset to obtain per-sample gradient signals. The subsequent subset selection step is negligible and is effectively identical across all methods, as it reduces to sorting or indexing the derived scores. For this reason, separating gradient-computation time and selection time would not meaningfully change the comparison, since the selection component contributes only a constant and uniformly small cost across all coreset methods.
> >
> > > **Model-based signals often provide richer information than embedding-space measures alone. It would be interesting to see whether the reported gains persist for larger models or fine-tuning scenarios, such as with SLMs, rather than being limited to smaller architectures like ResNets. Additionally, do the improvements translate to other modalities, for instance in text or large-scale training settings?**
> >
> > Our goal in this paper is to test whether a purely geometric, training-free objective can remain competitive in standard image classification without relying on training signals. Although we have not extended SubZeroCore to text domains, the method itself is domain-agnostic: it only requires a fixed embedding space and does not assume vision-specific structure. In text settings, SubZeroCore could operate on embeddings from large pretrained text encoders, and for large-scale vision, a similar embedding space can be obtained through self-supervised models such as DINO or CLIP. These scenarios are particularly appealing because training such embedding models from scratch is often infeasible due to hardware constraints, whereas SubZeroCore can operate directly on the available embeddings without additional training for deriving training signals and dynamics (unlike other compared coreset selection methods). We thank the reviewer for highlighting this benefit of SubZeroCore and will clarify this point in the revised version.
> >
> > > **For noisy outliers, the normalized scores across the radii distribution appear to favor dense clusters while penalizing dense outliers. Empirically, it would be helpful to know how this trade-off behaves with higher levels of label noise (e.g., >10%) [...].**
> >
> > Our density normalization is designed to suppress small, dense outlier clusters while still preserving genuine high-density regions, but we agree that its behavior under stronger label corruption warrants a more detailed analysis. We are currently running experiments with noise levels above 10%. Preliminary results show that the relative ordering of the methods remains consistent and that SubZeroCore does not collapse toward dense outlier clusters as noise increases. We will include the extended noise analysis in the supp. materials.
> >
> > > **The use of the K-NN metric may be sensitive to noisy data and outliers, which could affect the reliability of the evaluation, although authors argue the robustness to 10% label noise. A systematic ablation or analysis showing how the proposed normalized scores mitigate this sensitivity would help clarify the robustness of the method and strengthen the empirical claims.**
> >
> > While K-NN distances can be sensitive to noise, SubZeroCore mitigates this in two ways: First, the K-NN are computed in a fixed embedding space and do not use labels, so label noise only affects the downstream evaluation model rather than the geometry on which the density scores are defined. Second, we do not use raw K-NN radii directly: we normalize them with a Gaussian weighting around the empirical mean radius, which downweights both very large radii (isolated geometric outliers) and extremely small radii (tiny over-dense pockets), and places highest weight on average-density regions. This makes the normalized scores substantially less sensitive to noisy outliers than a plain K-NN heuristic.
> > The 10% noise experiment already shows improved robustness over vanilla facility location, and we will add further analysis to clarify this effect.
> >
> > > **Since the method reduces to a weighted Facility Location problem, it resembles a K-medoids variant. How does it compare to clustering algorithms such as BanditPAM and BanditPAM++?**
> >
> > Our objective is a weighted facility-location function. Although facility location resembles k-medoids when similarity is defined as negative distance, SubZeroCore is not a clustering method. Our K controls the density estimator, not the number of cluster centers. BanditPAM and BanditPAM++ explicitly optimize the k-medoids reconstruction loss, while SubZeroCore optimizes a submodular objective for supervised coreset selection. Because the goals and optimization criteria differ, these clustering methods are not directly comparable baselines.

---

> > > ### Author Response · Authors · 2025-11-15
> > >
> > > Based on the recommendations of Reviewer 2CD1, we also added Uniform Sampling (Random), Temporal Dual-Depth Scoring by Zhang et al. CVPR 2024, and ELFS by Zheng et al. ICLR 2025 to the main table. This results in the table:
> > >
> > > | Method | 99.9% | 99.5% | 99% | 95% | 90% | 80% | 70% | 60% | 50% | 40% | 10% | Train |
> > > | - | - | - | - | - | - | - | - | - | - | - | - | - |
> > > | Random | 21.0 | 30.8 | 36.7 | 62.5 | 75.7 | 87.1 | 90.2 | 92.1 | **93.3** | 94.0 | 95.2 | ✗ |
> > > |CCS (Grad)| 19.1 | 29.2 | 36.5 | 62.8 | 73.1 | 86.3 | 89.9 | 90.0 | 90.0 | 89.9 | 90.0 | ✓ |
> > > | ELFS | 13.7 | 20.9 | 25.3 | 39.7 | 52.7 | 76.8 | 89.2 | 91.7 | 91.9 | 92.3 | 92.6 | ✓ |
> > > | TDDS | 18.3 | 32.4 | 39.1 | 63.7 | 76.8 | 87.1 | 90.6 | 92.5 | 93.3 | 94.0 | 95.3 | ✓ |
> > > |**SubZeroCore**| **24.0** | **32.9** | **39.8** | **63.9** | **77.4** | **87.3** | **90.8** | **92.5** | 93.2 | **94.1** | **95.3** | ✗ |
> > >
> > > Comparison with InfoMax on CIFAR-10 (ResNet-18) remains unchanged:
> > >
> > > | Pruning (α) | 99.9% | 99.5% | 99% | 95% | 90% | 80% | 70% | 60% | 50% | 40% | 10% | Train |
> > > | - | - | - | - | - | - | - | - | - | - | - | - | - |
> > > | Random | 21.0 | 30.8 | 36.7 | 62.5 | 75.7 | 87.1 | 90.2 | 92.1 | **93.3** | 94.0 | 95.2 | ✗ |
> > > | Forgetting | 12.4 | 15.6 | 19.9 | 33.8 | 57.1 | 84.0 | 87.8 | 89.7 | 91.3 | 92.5 | 94.9 | ✓ |
> > > | EL2N | 11.0 | 10.7 | 12.5 | 23.5 | 55.6 | 86.5 | 88.9 | 90.5 | 91.5 | 92.4 | 95.0 | ✓ |
> > > | Entropy | 18.1 | 25.6 | 33.1 | 55.8 | 71.8 | 86.7 | 89.0 | 90.7 | 91.6 | 92.6 | 95.1 | ✓ |
> > > | **SubZeroCore (ours)** | **24.0** | **32.9** | **39.8** | **63.9** | **77.4** | **87.3** | **90.8** | **92.5** | 93.2 | **94.1** | **95.3** | ✗ |

---

> > > ### Author Response · Authors · 2025-11-20
> > > **Revised Manuscript and Supp. Materials**
> > >
> > > We uploaded the revised manuscript and added the suggested experiments as well as the suggested discussions. For the points raised by the reviewer, we refer to:
> > >
> > > - Table 2 & 3: Additional experiments with recent (and suggested) methods under the DeepCore benchmark setting.
> > > - Section 5, last paragraph: Discussion on other modalities.
> > > - Appendix L: Theoretical Analysis showing how the proposed normalized scores mitigate sensitivity.
> > > - Appendix K: Additional experiments under higher noise rates.
> > > - Appendix N: Discussion about the relation to other submodular objectives.

---

### Official Review · Reviewer_CYjx · 2025-11-03

**Soundness:** 2
**Presentation:** 3
**Contribution:** 3
**Rating:** 2
**Confidence:** 4

**Summary:**

SubZeroCore presents a training-free coreset selection method that optimally balances submodular coverage and density via a closed-form solution controlled by a single hyperparameter. Training-free coreset selection is desirable; many recent methods are reliant on training-based signals (e.g., gradients, forgetting scores). Empirical results show SubZeroCore matches or exceeds the performance of training-based and geometry-driven baselines on CIFAR-10 and ImageNet-1K, particularly at high pruning rates and in some settings with label noise.

**Strengths:**

- The method described is interesting and unifies competing goals into a single formulation. The ability to control the tradeoff between coverage and density by one hyperparameter is also compelling.

- The method derivation based on density-weighted to systematically downweighting potential noisy examples seems to lead to increased robustness to noise.

- The speed of the method is attractive.

- The results demonstrating strong performance at high prune rates are strong.

**Weaknesses:**

- Coreset selection is challenging whether or not one uses training signals. Interestingly, reports (e.g., Griffin et al. https://arxiv.org/pdf/2411.15349) have shown that uniform sampling is a very competitive baseline.  This paper has no such analysis against the competitive baseline.  EG. Looking at the prune rate of 70% for CIFAR10 in that paper and this paper (both using ResNet18 as the downstream detector), the uniform sampler performs within variance to this paper 90.61 +- .44 (AND the uniform sampler uses no class labels and no training).

- The paper defines the problem to be fully supervised setup with labels for all samples (tacitly described in Sec 2).  This differs from the original coreset definition Sener and Savarese 2017 active learning framework, where only some labels are available and the goal is to select the next subset to label. However, the actual method does not seem to use the class labels (although this is not clear because the notation shifts to only using a $\mathbf{X}$ to subsume the class labels.  So it is not clear whether or not the later derivations are performed per class or agnostic to class.  (While this shift is more common in recent coreset literature, it is problematic because it abandons the core motivation of Sener and Savarese: reducing labeling costs for large datasets.)

- Competitive methods are missing from the results.  e.g., Temporal Dual-Depth Scoring Zhang et al. CVPR 2024.  This example may use training, but it is still quite relevant. e.g.2: ELFS by Zheng et al. ICLR 2025 which does not use ANY labels.  No comparative result in the tables exist for papers after 2021?  This is concerning.  Furthermore, increasing the number of datasets used in the analysis would help the reader understand the method's behavior; datasets that have different comparative properties than just size like the two used here, such as gross class imbalance.

- Although the method is interesting, it seems like it may struggle for imbalanced scenarios.  Similarly, the paper suggests superior robustness to label noise, but this is only minimally analyzed in Fig. 5.  One expects more thorough analysis for a core property.

**Questions:**

- Clarify if and how the class labels are actually used in the method.  If they are not used the paper should more clearly state that.  EG one can have class labels but not use training.
- Why are no competitive results included from the recent literature? This is a significant weakness of the paper.
- How does the method perform under higher noise rates?
- HOw does the method perform under class imbalance?

**Details Of Ethics Concerns:**

Probably this is not a true ethics concern.  But, it is very clear to me that 13521 and 2361 were contributed by the same authors.  The problem statements are the same, chunks of text are common, and the results have the same limitations.  The methods seem to be sufficiently different (although in the end all they are really doing is fitting local densities to data-samples and the sampling from those densities).

---

> ### Author Response · Authors · 2025-11-15
>
> We are glad that the reviewer finds the unified formulation of coverage and density, and the ability to control their trade-off via a single hyperparameter, to be compelling. We also appreciate the recognition that our objective can systematically downweight noisy examples. In the following, we address the remaining concerns and clarify the points raised in the review.
>
> # Weaknesses
>
> > **[...] Griffin et al. have shown that uniform sampling is a very competitive baseline. [...] Competitive methods are missing from the results. e.g., Temporal Dual-Depth Scoring Zhang et al. CVPR 2024, [...] ELFS by Zheng et al. ICLR 2025**
>
> We appreciate the reviewer’s emphasis on strong baselines and recent methods.
> On the point of missing recent methods, we agree that this was important to address. In the updated tables, we have added uniform sampling (random), CCS when driven by gradient norms, the InfoMax and D2Pruning-based scores (Forgetting, EL2N, Entropy), as well as ELFS and TDDS. These additions cover training-based approaches proposed after 2021. Across these new comparisons, SubZeroCore remains competitive, particularly at aggressive pruning levels where geometric selection is most valuable.
>
> | Method | 99.9% | 99.5% | 99% | 95% | 90% | 80% | 70% | 60% | 50% | 40% | 10% | Train |
> | - | - | - | - | - | - | - | - | - | - | - | - | - |
> | Random | 21.0 | 30.8 | 36.7 | 62.5 | 75.7 | 87.1 | 90.2 | 92.1 | **93.3** | 94.0 | 95.2 | ✗ |
> |CCS (Grad)| 19.1 | 29.2 | 36.5 | 62.8 | 73.1 | 86.3 | 89.9 | 90.0 | 90.0 | 89.9 | 90.0 | ✓ |
> | ELFS | 13.7 | 20.9 | 25.3 | 39.7 | 52.7 | 76.8 | 89.2 | 91.7 | 91.9 | 92.3 | 92.6 | ✓ |
> | TDDS | 18.3 | 32.4 | 39.1 | 63.7 | 76.8 | 87.1 | 90.6 | 92.5 | 93.3 | 94.0 | 95.3 | ✓ |
> |**SubZeroCore**| **24.0** | **32.9** | **39.8** | **63.9** | **77.4** | **87.3** | **90.8** | **92.5** | 93.2 | **94.1** | **95.3** | ✗ |
>
> ### Comparison with InfoMax on CIFAR-10 (ResNet-18)
> | Pruning (α) | 99.9% | 99.5% | 99% | 95% | 90% | 80% | 70% | 60% | 50% | 40% | 10% | Train |
> | - | - | - | - | - | - | - | - | - | - | - | - | - |
> | Random | 21.0 | 30.8 | 36.7 | 62.5 | 75.7 | 87.1 | 90.2 | 92.1 | **93.3** | 94.0 | 95.2 | ✗ |
> | Forgetting | 12.4 | 15.6 | 19.9 | 33.8 | 57.1 | 84.0 | 87.8 | 89.7 | 91.3 | 92.5 | 94.9 | ✓ |
> | EL2N | 11.0 | 10.7 | 12.5 | 23.5 | 55.6 | 86.5 | 88.9 | 90.5 | 91.5 | 92.4 | 95.0 | ✓ |
> | Entropy | 18.1 | 25.6 | 33.1 | 55.8 | 71.8 | 86.7 | 89.0 | 90.7 | 91.6 | 92.6 | 95.1 | ✓ |
> | **SubZeroCore (ours)** | **24.0** | **32.9** | **39.8** | **63.9** | **77.4** | **87.3** | **90.8** | **92.5** | 93.2 | **94.1** | **95.3** | ✗ |
>
> > **[...] active learning framework [...]**
>
> We thank the reviewer for raising this distinction. Our setting follows the recent supervised coreset literature, where the goal is to prune an already labeled dataset rather than select points to be labeled, which is the focus of classical active learning.
> SubZeroCore does not use class labels when computing scores and therefore operates label-agnostically within this supervised pruning setup. We agree that the connection to the original active-learning formulation is worth clarifying, and we will add a short discussion in the supplementary material to explicitly contrast these two problem settings.
>
> # Questions
>
> >**Clarify if and how the class labels are actually used in the method. [...]**
>
> We appreciate the request for clarification. SubZeroCore does not use class labels to compute scores, but we apply the selection procedure class-wise for efficiency. Labels are therefore used only to partition the data, not to drive the scoring or optimization. We will clarify this distinction in the revised version.
>
> > **How does the method perform under higher noise rates?**
>
> We are currently evaluating noise levels beyond 10%. Early results indicate that the relative ranking of methods is stable and that SubZeroCore does not drift toward dense outlier clusters as noise increases. We will include the extended noise analysis in the supplementary material.
>
> > **How does the method perform under class imbalance?**
>
> SubZeroCore is applied class-wise (in contrast to training signal-based coreset selection methods), so each class is allocated its own quota before selection. This makes the method naturally robust to class imbalance, since rare classes are not overwhelmed by majority classes during scoring or selection (which alters the training dynamics). We will clarify this behavior in the revised version.

---

> > ### Author Response · Authors · 2025-11-20
> > **Revised Manuscript and Supp. Materials**
> >
> > We uploaded the revised manuscript and added the suggested experiments as well as the suggested discussions. For the points raised by the reviewer, we refer to:
> >
> > - Section 3.6: Clarification of how class labels are used and how SubZeroCore performs under class imbalance.
> > - Table 2 & 3: Additional experiments with recent (and suggested) methods under the DeepCore benchmark setting.
> > - Appendix K: Additional experiments under higher noise rates
> > - Appendix M: Discussion about SubZeroCore vs Active Learning

---

### Meta-Review · Area_Chair_3BwJ · 2026-01-12

**Summary:**

SubZeroCore proposes a training-free coreset selection approach that unifies coverage and diversity based approaches into a single Facility Location style objective. The method operates purely on frozen embeddings, avoids gradients or training dynamics, and aims to be computationally efficient while remaining competitive.
 Overall, the approach and results seem correct. The major criticism from all the reviewers (and I agree for the most part), is aspects around the novelty, and comparison to baselines. A few reviewers have identified several missing baselines, as well as discussion and comparison to other coverage/diversity functions and even the Combinatorial Mutual Information functions. In addition, I would have loved to see some reasoning behind why the Facility Location makes sense here, or connect it to convergence bounds or generalization bounds like how it was done in CRAIG and GradMatch.

**Reviewer Concerns:**

Some of the reviewer concerns were addressed. Authors added Random (uniform), CCS(Grad), ELFS, TDDS, plus InfoMax-style scores (Forgetting/EL2N/Entropy) comparisons and updated tables. The authors also clarified issues around the label usage ambiguity (scores are label-agnostic) and higher label noise levels. The reviewers did bring an ethics related issue - the authors agree of another submission by the same group and there is shared/common language used in the motivation, etc. I would encourage the authors to please fix that though that is not a blatant ethics issue in my opinion, but there will likely be a separate ethics review.

**Reviewer Scores:**

I do not believe the reviewer scores would have significantly changed since many of the issues around novelty, baselines (though partially addressed) still remain. I would encourage the authors to address the several issues brought up for the next version of the paper. Ther approach and results are definitely promising - I would encourage the authors to address all issues and resubmit.

---

### Decision · Program_Chairs · 2026-01-26

Reject